# The MAST kinase KIN-4 carries out mitotic entry functions of Greatwall in *C. elegans*

Ludivine Roumbo [ID] [1,2], Batool Ossareh-Nazari [ID] [1,2], Suzanne Vigneron[3], Ioanna Stefani [ID] [1,2,4], Lucie Van Hove [ID] [1,2], Véronique Legros [ID] [1], Guillaume Chevreux[1], Benjamin Lacroix [ID] [3], Anna Castro [ID] [3], Nicolas Joly [ID] [1,2], Thierry Lorca [ID] [3] & Lionel Pintard [ID] [1,2 ✉]

## Abstract

**MAST-like, or Greatwall (Gwl), an atypical protein kinase related to the evolutionarily conserved MAST kinase family, is crucial for cell cycle control during mitotic entry. Mechanistically, Greatwall is activated by Cyclin B-Cdk1 phosphorylation of a 550 amino acids-long insertion in its atypical activation segment. Subsequently, Gwl phosphorylates Endosulfine and Arpp19 to convert them into inhibitors of PP2A-B55 phosphatase, thereby preventing early dephosphorylation of M-phase targets of Cyclin B-Cdk1. Here, searching for an elusive Gwl-like activity in *C. elegans*, we show that the single worm MAST kinase, KIN-4, fulfills this function in worms and can functionally replace Greatwall in the heterologous *Xenopus* system. Compared to Greatwall, the short activation segment of KIN-4 lacks a phosphorylation site, and KIN-4 is active even when produced in *E. coli*. We also show that a balance between Cyclin B-Cdk1 and PP2A-B55 activity, regulated by KIN-4, is essential to ensure asynchronous cell divisions in the early worm embryo. These findings resolve a long-standing puzzle related to the supposed absence of a Greatwall pathway in *C. elegans*, and highlight a novel aspect of PP2A-B55 regulation by MAST kinases.**

**Keywords** Development; Mitosis; Kinase; Phosphatase; *C. elegans*
**Subject Category** Cell Cycle

## Introduction

Entry into mitosis is triggered by a kinase cascade culminating with the abrupt activation of the Cyclin B-Cdk1 kinase, which phosphorylates numerous substrates to orchestrate the cellular reorganization required for chromosome segregation. The stable phosphorylation of these substrates critically requires the inhibition of the counteracting phosphatase, PP2A-associated with the regulatory subunit B55 (PP2A-B55) (Castilho et al, 2009; Vigneron et al, 2009; Mochida et al, 2016). To this end, Cyclin B-Cdk1 activates a positive feedback loop that involves the evolutionarily conserved MAST-L/Greatwall (hereafter Gwl) kinase (Fig. 1A) (Yu et al, 2006; Goldberg, 2010), initially described in flies as the *scant* (*Scott of the Antarctic*) mutation (White-Cooper et al, 1996; Glover, 2012). Gwl is a member of the MAST1-4 (microtubule associated serine/threonine) subfamily of AGC kinases (Manning et al, 2002). MAST kinases are widely expressed, particularly in the brain, and have been implicated in diverse biological processes, but little is known about their regulation (Rumpf et al, 2023). Most kinases are activated by phosphorylation of a conserved residue of the activation segment, but MAST kinases lack a phosphorylatable residue in that region (Somale et al, 2020). In contrast to MAST kinases, Gwl contains a unique and atypical long insertion of about 550 non-conserved amino acids in the activation segment. Phosphorylation of several residues in that region by Cyclin B-Cdk1 activates Gwl, which phosphorylates and activates the small alpha Endosulfine protein (Endos) and its close relative cyclic adenosine monophosphate–regulated phosphoprotein 19 (Arpp19) on the evolutionarily conserved DSG sequence motif (Gharbi-Ayachi et al, 2010; Mochida et al, 2010; Rangone et al, 2011; Vigneron et al, 2011; Kim et al, 2012). Then, phospho-Arpp19/Endos binds and inhibits PP2A-B55 (Gharbi-Ayachi et al, 2010; Mochida et al, 2010). Phosphorylated Arpp19/Endos displays high affinity to PP2A-B55 and a slow dephosphorylation rate, thus acting as a competitor of PP2A-B55 substrates (Williams et al, 2014; Labbé et al, 2021). How the regulation and cross-talk between Cyclin B-Cdk1 and the PP2A-B55 phosphatase ensure correct cell cycle length during development remains poorly understood.

With its highly stereotypical asymmetric and asynchronous cell divisions, the early *C. elegans* embryo provides an ideal system to study the mechanisms regulating cell cycle length during development with high spatiotemporal resolution (Tavernier et al, 2015; Pintard and Bowerman, 2019). The first embryonic division is asymmetric and generates two blastomeres of different sizes and developmental potentials that divide asynchronously (Brauchle et al, 2003; Tavernier et al, 2015). The anterior somatic AB blastomere enters mitosis roughly two minutes before the posterior germline $P_1$ blastomere (Deppe et al, 1978; Sulston et al, 1983).

[1]Université Paris cité, CNRS, Institut Jacques Monod, F-75013 Paris, France. [2]Programme Equipe Labellisée Ligue contre le Cancer, Paris, France. [3]Université de Montpellier, Centre de Recherche en Biologie Cellulaire de Montpellier, CNRS UMR 5237, 34293 Montpellier, Cedex 5, France. [4]Present address: Institute for Integrative Biology of the Cell, Commissariat à l'Énergie Atomique et Aux Énergies Alternatives, Centre National de la Recherche Scientifique, Université Paris-Saclay, 91190 Gif-sur-Yvette, France. ✉E-mail: lionel.pintard@ijm.fr

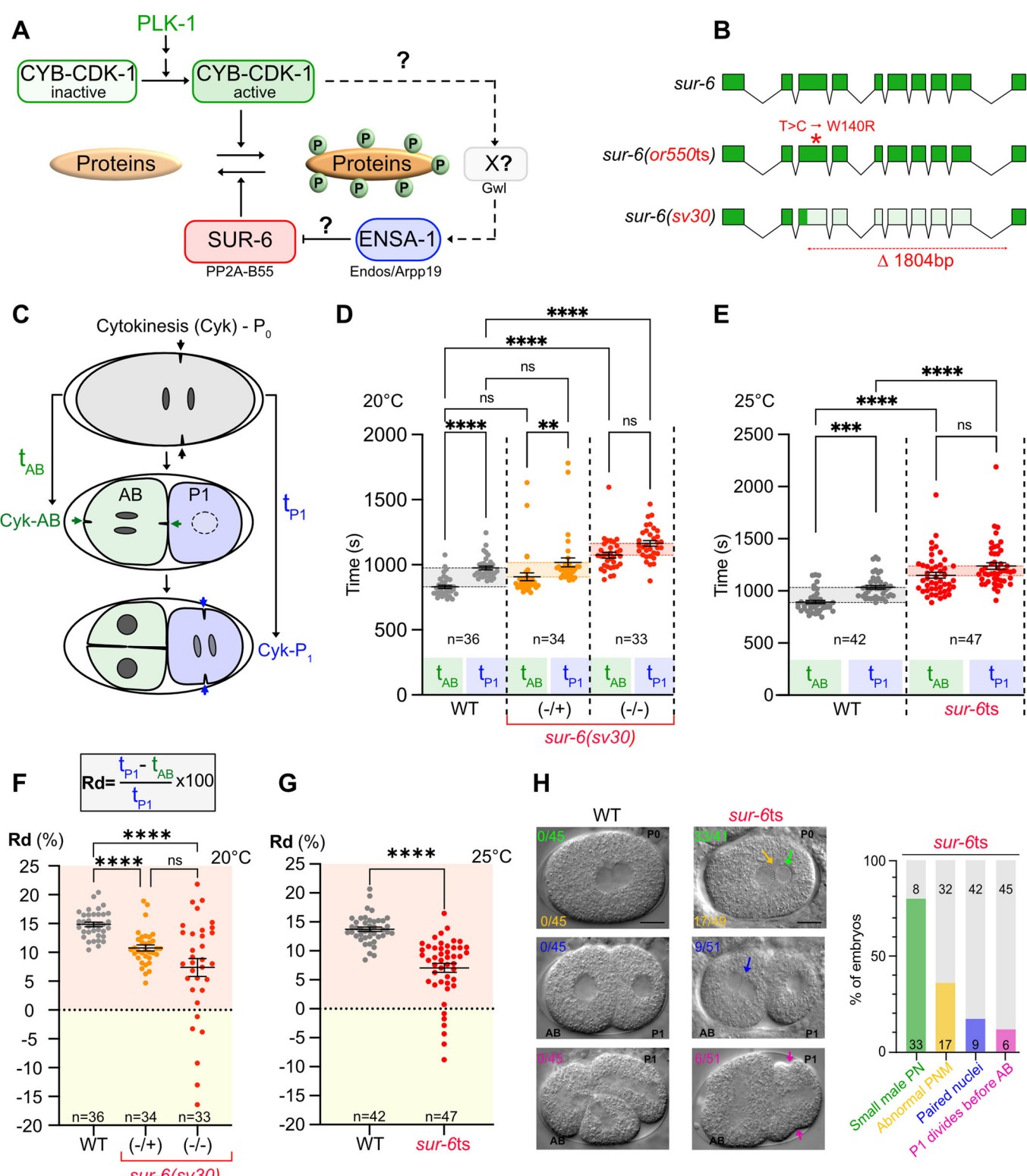

The earlier Cyclin B-CDK-1 activation in the AB blastomere is due to the enrichment of its activators, the Polo-like kinase-1 (PLK-1) and the CDC-25 phosphatase in this blastomere (Budirahardja and Gonczy, 2008; Nishi et al, 2008; Rivers et al, 2008; Tavernier et al, 2015). However, the role and regulation of PP2A-B55, which counteracts the action of Cyclin B-CDK-1, remain poorly understood in *C. elegans* (Fig. 1A). In this organism, the B55 subunit is encoded by the essential *sur-6* (SUppressor of activated *let-60* Ras) gene, initially identified in a genetic screen aimed at identifying regulators of the ras pathway during vulval development. Loss of *sur-6* function also causes

◄ **Figure 1.   The SUR-6$^{PP2A-B55}$ phosphatase regulates cell cycle asynchrony in 2-cell stage *C. elegans* embryos.**

(A) Schematic representation of the putative SUR-6$^{PP2A-B55}$ regulatory pathways in *C. elegans*. The *C. elegans* genome does not encode a Greatwall kinase (Gwl), but it encodes a homolog of Endos/Arpp19, known as ENSA-1. However, whether ENSA-1 inhibits SUR-6$^{PP2A-B55}$ activity is unknown. (B) *sur-6* gene structure (top). The *sur-6*ts allele, which harbors a point mutation in the third exon, encodes a mutated SUR-6 protein where an Arginine substitutes a conserved Tryptophan at position 140. The *sur-6(sv30)* deletion allele carries an 1804 bp deletion that deletes part of exons 3 to exon 10. (C) Schematic of the first and second cell divisions of *C. elegans* embryos. The arrows show the ingression of the cytokinetic furrows in $P_0$, AB, and $P_1$ blastomeres. The dashed circle line in $P_1$ shows the nuclear envelope breakdown. In wild-type embryos, the anterior AB blastomere (light green) divides roughly two minutes before the posterior $P_1$ blastomere (light blue). $t_{AB}$ and $t_{P1}$ is the elapsed time between $P_0$-AB, and $P_0$-$P_1$ cytokinetic furrow initiation. The relative difference (Rd) in AB and $P_1$ cell cycle lengths corresponds to [($t_{AB}$-$t_{P1}$)/$t_{P1}$x100]. (D, E) Graph presenting the cell cycle length of AB (light green) and $P_1$ (light blue) blastomeres in embryos of the indicated genotype represented as the mean ± standard error to the mean. *sur-6(sv30)* heterozygous and homozygous mutants are denoted (-/+) and (-/-) respectively. We could not film *sur-6(-/-)* homozygous mutant embryos at 25 °C as they cannot divide at this temperature, presenting major defects, and we had to record these embryos at 20 °C. *sur-6*ts embryos were recorded at 25 °C. n = number of embryos analyzed. Non-parametric tests (Kruskal–Wallis) were used to calculate *p* values, which are displayed as follows: ns = $p > 0.05$; * = $p < 0.05$; ** = $p < 0.01$; *** = $p < 0.001$; **** = $p < 0.0001$. Exact *p* values from (D) (L-R); $p < 0.0001$, $p < 0.0001$, $p = 0.0018$, $p < 0.0001$. Exact *p* values from (E) (L-R) $p = 0.0004$, $p < 0.0001$, $p = 0.0002$. Error bars display the standard error to the mean. ns no-significant differences. (F, G) Graph presenting the relative difference of AB and $P_1$ cell cycle lengths in embryos of the indicated genotypes in percentage, represented as the mean ± standard error to the mean. n number of embryos analyzed. *sur-6(sv30)* heterozygous and homozygous mutants are denoted (-/+) and (-/-) respectively. Non-parametric tests (Kruskal–Wallis) were used to calculate *p* values, which are displayed as follows: ns = $p > 0.05$; **** = $p < 0.0001$. Exact *p* values from (F) (L-R); $p < 0.0001$, $p < 0.0001$. Exact *p* values from (G) (L-R) $p < 0.0001$. Error bars display the standard error to the mean. ns no-significant differences. (H) DIC micrographs of WT and *sur-6*ts embryos during the first division and at the two-cell stage. Compared to the wild-type, the male pronucleus is smaller (green arrow), pronuclear meeting (PNM) is abnormal in *sur-6*ts embryos (yellow arrow). At the two-cell stage, *sur-6*ts embryos present paired nuclei phenotype (blue arrow) and defective AB-$P_1$ asynchrony of division, with $P_1$ eventually dividing before AB (magenta arrows). The fraction of embryos that showed the phenotype is indicated at the top left of each image. The anterior end of the embryo is to the left in this and other figures. Scale bar: 10 µm. The graph presents the percentage of *sur-6*ts embryos with small paternal pronucleus (PN), abnormal pronuclei meeting (PNM), paired nuclei at the two-cell stage, and inverted asynchrony with $P_1$ eventually dividing before AB. The graph indicates the number of embryos analyzed and generated by aggregation over more than three independent experiments. Source data are available online for this figure.

cell cycle defects and embryonic lethality (Sieburth et al, 1999; Kao et al, 2004; O'Rourke et al, 2011). Notably, in *sur-6* null or temperature-sensitive mutant embryos, the posterior $P_1$ blastomere sometimes divides before the AB blastomere (Kao et al, 2004; O'Rourke et al, 2011), suggesting that SUR-6$^{PP2A-B55}$ may contribute to the robust and reproducible cell cycle asynchrony between AB and $P_1$ blastomeres. Beyond cell cycle timing regulation, SUR-6$^{PP2A-B55}$ regulates pronuclear sizes in the one-cell embryo (i.e the male pronucleus is smaller than the female pronucleus in *sur-6* mutants), centrosome separation and maturation during mitosis (Kitagawa et al, 2011; Song et al, 2011; Boudreau et al, 2019) and lamina depolymerization (Kapoor et al, 2023).

Despite its critical role in regulating cell division and development, how PP2A-B55$^{SUR-6}$ is regulated is not understood in *C. elegans* (Fig. 1A). No homolog for Gwl, with its peculiar and atypical activation loop, has been identified in worms (Kim et al, 2012), raising questions about the mechanism by which Cyclin B-CDK-1 phosphorylated substrates accumulate during mitosis in *C. elegans*.

Paradoxically, nematodes do encode other components of the pathway, including a single Endos protein called ENSA-1 (ENdoSulphine Alpha) that retains the highly conserved sequence motif (DSG), phosphorylated by Gwl in flies and vertebrates (Kim et al, 2012) but whether ENSA-1 is functional in worms is elusive (Fig. 1A).

Here, we identify and functionally characterize a new pathway that inhibits SUR-6$^{PP2A-B55}$ activity in *C. elegans* embryos. In this pathway, the single worm-encoded MAST kinase KIN-4 (KINase 4), previously associated with aging and thermotaxis phenotypes (An et al, 2019; Nakano et al, 2022), directly phosphorylates ENSA-1 on the canonical DSG motif, in vitro and in vivo, and converts it into a potent SUR-6$^{PP2A-B55}$ inhibitor. Remarkably, while this pathway is not essential for survival, its inactivation rescues the embryonic lethality and cell cycle timing defects associated with decreased SUR-6$^{PP2A-B55}$ function. Furthermore, we show that KIN-4 fully rescues the depletion of Greatwall in *Xenopus* egg extracts, indicating it is a bona fide functional Greatwall homolog in worms. These findings solve a long-standing paradox regarding the absence

of a Greatwall-like pathway in *C. elegans* and provide new insights into PP2A-B55 regulation by MAST kinases with potential implications in several cellular and biological processes.

## Results

### Loss of *sur-6$^{PP2A-B55}$* function affects AB-$P_1$ cell cycle asynchrony in two-cell *C. elegans* embryos

The early *C. elegans* embryo alternates between rapid DNA replication and mitotic phases, with no intervening gap phases (Sulston et al, 1983). In wild-type two-cell stage embryos, the anterior somatic AB blastomere undergoes mitosis roughly two minutes before the posterior germline blastomere $P_1$ (Brauchle et al, 2003; Tavernier et al, 2015). Two earlier studies reported cell cycle timing defects in *sur-6(sv30)$^{PP2A-B55}$* null or *sur-6(or550)$^{PP2A-B55}$* temperature-sensitive (hereafter *sur-6*ts) mutant embryos, where the $P_1$ blastomere sometimes divided before the AB blastomere (Kao et al, 2004; O'Rourke et al, 2011). However, only a few embryos were recorded and analyzed in these two studies, and neither study quantified the cell cycle lengths of AB and $P_1$, making it unclear whether, in these *sur-6* mutants, the $P_1$ blastomere was dividing faster or the AB blastomere was specifically delayed.

We reexamined this issue using time-lapse differential interference contrast (DIC) microscopy. We quantified the cell cycle length of AB and $P_1$ blastomeres in embryos expressing one *sur-6* copy (sv30 null heterozygous), *sur-6* null (sv30 homozygous), or the *sur-6*ts allele (Fig. 1B). For *sur-6*ts mutants, we recorded embryos shifted 5-6 h at 25 °C. In contrast, for *sur-6(sv30)* mutants, we had to record embryos at 20 °C because they did not divide at 25 °C. We measured the elapsed time between the initiation of cytokinesis (furrow formation, arrows as shown in Fig. 1C) in $P_0$, AB, and $P_1$ blastomeres to obtain the cell cycle lengths of AB and $P_1$ ($t_{AB}$ and $t_P$) respectively (Fig. 1D,E). We then calculated the relative difference (Rd) between the division timing of AB and $P_1$ blastomeres (Rd = [($t_{P1}$- $t_{AB}$)/$t_{P1}$ X 100]). When the AB

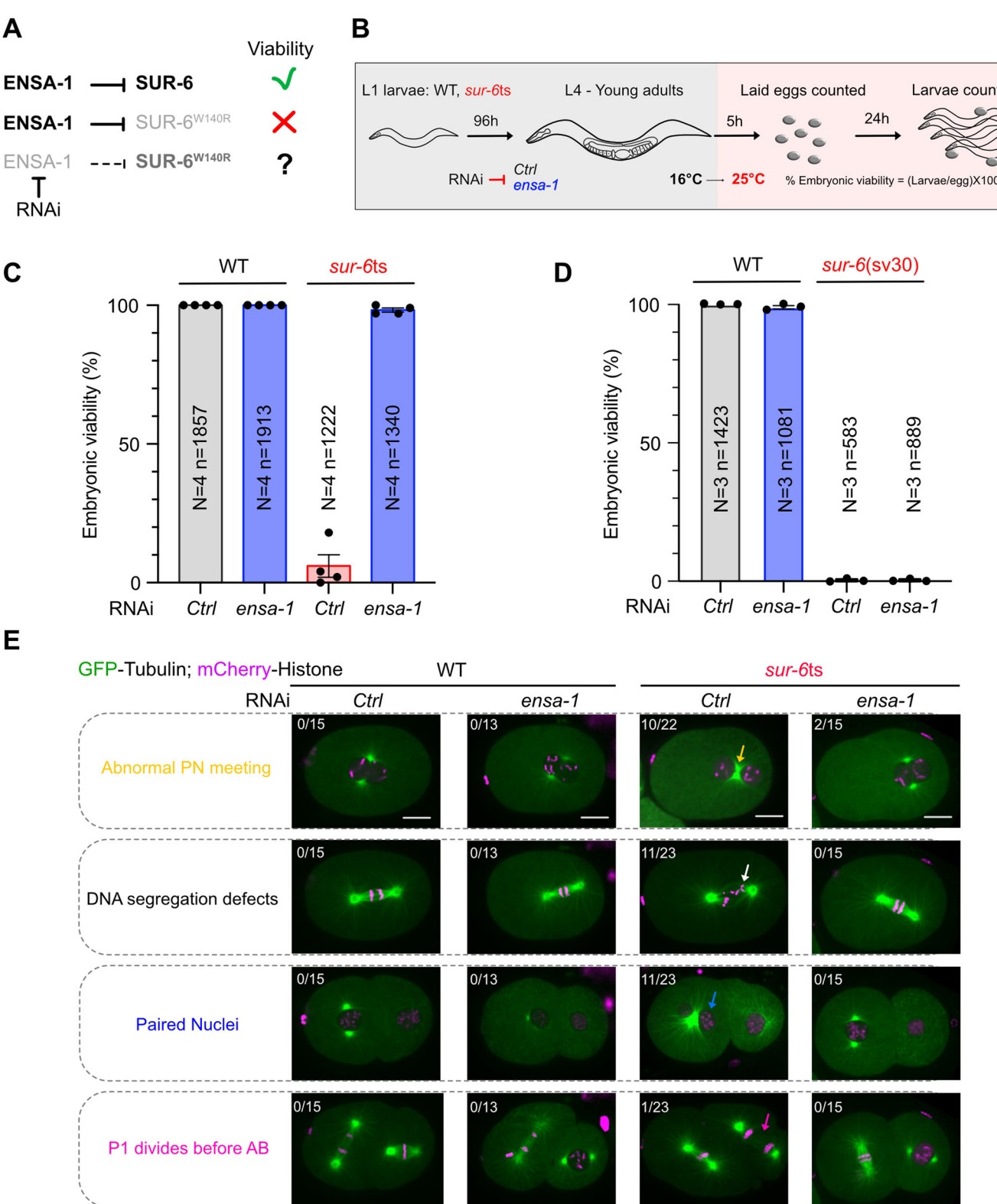

blastomere divides before $P_1$, as in wild-type, the relative difference (Rd) is positive, it equals zero when AB and $P_1$ divide synchronously, and is negative when $P_1$ divides before AB (Fig. 1F,G).

In wild-type two-cell embryos, the cell cycle length of AB is shorter than that of $P_1$, with $P_1$ dividing roughly 120 s after AB

(Fig. 1D,E). Consequently, Rd = 15, indicating that the AB cell cycle is ~15% faster than $P_1$ (Fig. 1F,G). Previous work showed that when the mitotic PLK-1 or CDK-1 kinases are partially inactivated, both AB and $P_1$ divisions are delayed. However, the cell cycle of the $P_1$ blastomere is comparatively more delayed than AB,

◄ **Figure 2.** *ensa-1 inactivation suppresses partial loss-of-sur-6*$^{PP2A-B55}$ *function.*

(A) Schematic of the rationale for testing if ENSA-1 is a SUR-6$^{PP2A-B55}$ inhibitor. If ENSA-1 acts as a SUR-6 inhibitor in normal conditions, it should become toxic when SUR-6 activity is compromised (light gray). Reducing ENSA-1 function by RNAi (removing the inhibitory effect) should compensate for and rescue SUR-6 function and embryo viability. (B) Schematic of the approach to test whether *ensa-1* inactivation by RNAi restores the embryonic viability of the *sur-6*ts mutant. L1 larvae WT or *sur-6*ts were grown on control or *ensa-1* RNAi plates until adulthood. Then, animals were shifted for 5 h at 25 °C before a progeny test was performed to determine the percentage of viability in each condition. (C, D) Graphs showing the percentage of embryonic viability, represented as the mean ± standard error to the mean, of wild-type N2 and *sur-6*ts, exposed to control or *ensa-1* RNAi after shifting the animals 5 h at the restrictive temperature (25 °C). The experiment with *sur-6(sv30)* null was performed at 20 °C. *N* is the number of independent experiments, and *n* is the total number of embryos counted. (E) Images from a time-lapse spinning disk confocal movie of wild-type and *sur-6*ts embryos expressing GFP::Tubulin (green) and mCherry-HIS-11 (magenta) exposed to control or *ensa-1(RNAi)*. The arrows point to the mispositioned and unseparated centrosomes leading to the abnormal pronuclear meeting (yellow), the DNA segregation defects (white arrow), paired nuclei phenotype at the two-cell stage (blue arrow), with P1 dividing before AB (magenta arrow). The fraction of embryos that showed the phenotype is indicated at the top left of each image. Scale bar: 10 µm. Source data are available online for this figure.

increasing Rd roughly to 25% (Budirahardja and Gonczy, 2008; Noatynska et al, 2010).

In *sur-6(sv30)/+* heterozygous mutant embryos, we found that the cell cycle length of AB and $P_1$ was extended (Fig. 1D). However, in this case, the AB blastomere was more affected, resulting in a significant decrease in the relative difference between AB and $P_1$ cell cycle length. The AB blastomere was only 10% faster than $P_1$ in these embryos (Fig. 1F). We observed the same phenotype in *sur-6(sv30)* homozygous mutants, but in some embryos, $P_1$ even divided before AB (Fig. 1D–F).

We repeated this analysis with *sur-6*ts mutant embryos (Fig. 1E). Both the AB and $P_1$ cell cycle lengths were prolonged in this mutant, and again, as in *sur-6(sv30)* embryos, AB was significantly more affected than $P_1$, resulting in a significant decrease of Rd and in several embryos, $P_1$ even divided before AB (Rd < 0) (Fig. 1G).

To determine which cell cycle phase was prolonged in the *sur-6* mutants, we quantified the length of interphase (time between furrow ingression in $P_0$ and nuclear envelope permeabilization in AB and $P_1$) and mitosis (time between nuclear envelope permeabilization and furrow ingression in AB and P1 blastomeres), respectively, in WT and *sur-6* mutants (Figs. 1 and EV1A). The interphase of both AB and $P_1$ blastomere was prolonged. However, only the length of mitosis of the AB blastomere was prolonged in *sur-6* mutants, indicating that the AB blastomere is more sensitive to PP2A-B55 inactivation than the $P_1$ blastomere (Figs. 1 and EV1B,C). Therefore, the decrease in AB-P1 cell asynchrony observed in *sur-6* mutants mainly results from extended mitosis of the AB blastomere.

Beyond these cell cycle timing defects, we observed other phenotypes in *sur-6*ts mutant embryos, including smaller male pronucleus (PN), defects in pronuclei meeting (PNM), and paired nuclei phenotypes at the two-cell stage (Fig. 1H), in line with previous reports (O'Rourke et al, 2011; Boudreau et al, 2019; Kapoor et al, 2023).

Thus, the phenotypes resulting from *sur-6* inactivation (AB division delayed) indicate that SUR-6$^{PP2A-B55}$ activity is critically required to regulate mitotic progression and the AB-$P_1$ cell division asynchrony. The delayed AB division is the opposite of what one finds upon inactivation of the CDK-1 or the PLK-1 kinase (where the $P_1$ division is delayed) (Budirahardja and Gonczy, 2008; Noatynska et al, 2010). These observations indicate that a tight balance between Cyclin B-Cdk1 and its counteracting SUR-6$^{PP2A-B55}$ activity is essential to regulate and ensure robust cell cycle asynchrony during embryonic development.

## The single Endosulfine ENSA-1 is a SUR-6$^{PP2A-B55}$ inhibitor in *C. elegans*

We next asked whether ENSA-1 is functional in worms and regulates SUR-6$^{PP2A-B55}$ activity, as it does in other metazoans. If ENSA-1 is a bona fide SUR-6$^{PP2A-B55}$ inhibitor, its depletion by RNAi should increase SUR-6$^{PP2A-B55}$ activity and thus suppress the lethality associated with the reduced activity of the *sur-6*ts mutant (O'Rourke et al, 2011) (Fig. 2A,B). Consistently, while *ensa-1* inactivation by RNAi did not affect the viability of wild-type N2 worms, it robustly suppressed the embryonic lethality associated with the *sur-6*ts allele, as more than 90% of *ensa-1(RNAi); sur-6*ts embryos were viable (Fig. 2C). This suppression requires residual SUR-6 activity because *ensa-1* depletion did not affect the lethality of homozygous *sur-6(sv30)* null mutant embryos (Fig. 2D). These results suggest that ENSA-1 directly acts on SUR-6 and inhibits PP2A-B55 activity, as reported previously for Arpp19 and Ensa in other systems (Gharbi-Ayachi et al, 2010; Mochida et al, 2010; Labbé et al, 2021; Padi et al, 2024).

We then asked whether *ensa-1* inactivation suppresses the other phenotypes of *sur-6*ts embryos. To this end, we generated a *sur-6*ts line expressing Histone and Tubulin (TBA-2) fused to mCherry and GFP, respectively, to analyze the embryos using spinning disk confocal microscopy. Here, we shifted *sur-6*ts animals from the L1 stage to young adults for at least 48 h at the restrictive temperature of 25 °C. Remarkably, the *sur-6*ts phenotypes, including the defects in pronuclear meeting due to defective centrosome separation, defects in chromosome segregation, paired nuclei phenotype, and aberrant asynchrony of division at the two-cell stage, with the $P_1$ blastomere eventually dividing before AB, were suppressed by the concomitant inactivation of *ensa-1* by RNAi (Fig. 2E).

Taken together, these observations indicate that, although not essential for viability, ENSA-1 is a potent SUR-6$^{PP2A-B55}$ inhibitor, detrimental to embryonic viability when SUR-6$^{PP2A-B55}$ activity is compromised.

## ENSA-1 is phosphorylated at the evolutionarily conserved DSG motif in early *C. elegans* embryos

Next, we investigated the mechanisms by which ENSA-1 inhibits SUR-6$^{PP2A-B55}$ activity. Through phosphopeptide enrichments from embryo extracts coupled with mass spectrometry (Fig. 3A), we found that ENSA-1 is phosphorylated at multiple sites, including five S/T-P sites, matching the consensus for Cyclin-Cdk1 (Fig. 3B; Table EV1), and at the DSG sequence motif (S61) (Fig. EV2A,B), which in other

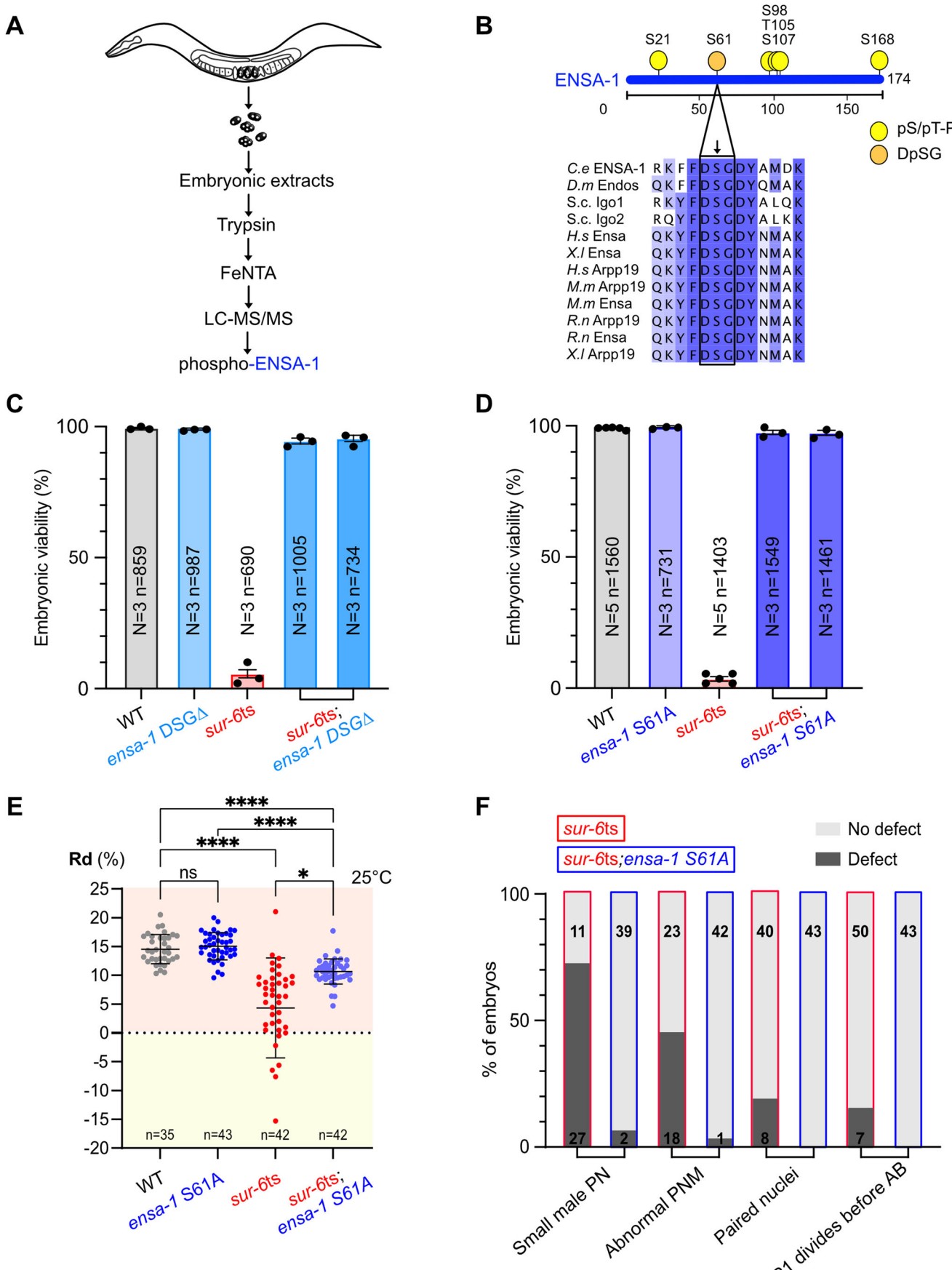

◄ **Figure 3. ENSA-1 phosphorylation at the DSG sequence motif is required for SUR-6$^{PP2A-B55}$ inhibition.**

(A) Flowchart of the approach to delineate the *C. elegans* phosphoproteome. Embryonic extracts prepared from young adults were digested with trypsin, and phosphopeptides were affinity purified using Fe-NTA-immobilized metal ion affinity chromatography (IMAC) columns before identification by tandem mass spectrometry (LC-MS/MS). Multiple phosphosites were identified on ENSA-1. (B) Schematic representation of ENSA-1 and localization of the phosphorylation sites identified by tandem mass spectrometry from embryonic extracts (top panel). The orange circle indicates the phosphorylated residue at the DSG sequence motif, while the yellow circles indicate the phosphorylated residues matching the S/T-P consensus. The DSG sequence motif is conserved in *C. elegans* ENSA-1 (bottom panel). Multiple protein sequence alignments of the DSG sequence motif of Endosulfine and Arpp19 from several species. C. e. ENSA-1 (Q9XU56), D. m. Endosulfine (Q9VUB8), S. c. Igo1 (P53897) and Igo2 (Q9P305), H. s. Ensa (O43768), X. l. Ensa (Q7ZXH9), H. s. Arpp19 (P56211), M. m. Arpp19 (P56212), M. m. Ensa (P60840), R. n. Arpp19 (Q712U5), R. n. Ensa (P60841), X. l. Arpp19 (Q6DEB4). (H.s. *Homo sapiens*, X. l., *Xenopus laevis*, M. m. *Mus musculus*, R. n., *Rattus Norvegicus*, D. m. *Drosophila melanogaster*, C. e. *Caenorhabditis elegans*, S. c. *Saccharomyces cerevisiae*). (C) Graph showing the percentage of embryonic viability, represented as the mean ± standard error to the mean, of wild-type N2, *sur-6*ts, *ensa-1* DSGΔ *(tm2810)* and *sur-6*ts; *ensa-1* DSGΔ *(tm2810)* double mutants (two clones). N is the number of different experiments, and *n* is the total number of embryos counted. (D) Graph showing the percentage of embryonic viability, represented as the mean ± standard error to the mean, of wild-type N2, *ensa-1(S61A)*, *sur-6*ts, and *sur-6*ts; *ensa-1(S61A)* double mutants (two clones) cultivated at 16 °C from the L1 stage and shifted 5 h at 25 °C at adulthood. N is the number of independent experiments, and *n* is the total number of embryos counted. (E) Graph presenting the relative difference of AB and P$_1$ cell cycle lengths in embryos of the indicated genotypes in percentage, represented as the mean ± standard error to the mean. *n* number of embryos analyzed. Non-parametric tests (Kruskal–Wallis) were used to calculate *p* values, which are displayed as follows: ns = *p* > 0.05; * = *p* < 0.05; ** = *p* < 0.01; *** = *p* < 0.001; **** = *p* < 0.0001. Exact *p* values from (E) (L-R); *p* < 0.0001, *p* < 0.0001, *p* < 0.0001, *p* = 0.0121. Error bars display the standard error to the mean. ns no-significant differences. (F) Graphs presenting the percentage of single *sur-6*ts and double *sur-6*ts; *ensa-1(S61A)* mutant embryos showing the indicated phenotypes. Source data are available online for this figure.

metazoans is phosphorylated by the Gwl kinase, such that phospho-Endos/Arpp19 thereby becomes a potent PP2A-B55 inhibitor (Gharbi-Ayachi et al, 2010; Mochida et al, 2010; Padi et al, 2024).

To determine whether DSG motif phosphorylation is required for ENSA-1 inhibition of SUR-6, we used CRISPR/Cas9 to generate two double mutants. In the first, we introduced the SUR-6 W140R temperature-sensitive mutation into the strain harboring the *ensa-1(tm2810)* allele, which expresses ENSA-1 lacking 23 amino acids that include the conserved residues surrounding the DSG sequence motif (hereafter *ensa-1* DSGΔ) (Kim et al, 2012) (Fig. EV2A). In the second, we sequentially introduced the ENSA-1 S61A mutation and then the SUR-6 W140R mutation.

In contrast to the simple *sur-6*ts mutant, which produced dead embryos at restrictive temperature, *ensa-1(DSGΔ)*; *sur-6*ts and *ensa-1* S61A; *sur-6*ts double mutants produced viable embryos (Fig. 3C,D). We also found that ENSA-1 S61A mutation partially restored regular AB and P$_1$ cell cycle length and asynchrony to the *sur-6*ts mutant (Figs. 3E and EV2C) and, in addition, as expected, also suppressed the other phenotypes associated with the *sur-6*ts mutation (Fig. 3F). Taken together, these observations indicate that ENSA-1 phosphorylation at the DSG motif is critically required for SUR-6$^{PP2A-B55}$ inhibition.

## The MAST kinase KIN-4 acts as a genetic suppressor of *sur-6*

Given that ENSA-1 is phosphorylated at the canonical DSG sequence motif in early embryos, a phosphorylation event is required for SUR-6$^{PP2A-B55}$ inhibition (Fig. 3); there must be at least one kinase in the worm genome with this activity.

However, as discussed earlier, the *C. elegans* genome does not encode a recognizable homolog of Gwl kinase (Kim et al, 2012), which typically phosphorylates Endos and Arpp19 at this site in humans, *Xenopus*, or *Drosophila*.

Gwl, also called MAST-L is a distantly related member of the MAST family of AGC kinases. This family comprises four classes (MAST1-4) that contain three conserved domains: a domain of unknown function DUF1908, a Serine/Threonine kinase domain related to AGC kinases (Pearce et al, 2010), and a C-terminal PDZ domain (Pearce et al, 2010; Rumpf et al, 2023) (Fig. 4A). It is

noteworthy that MAST3 reportedly phosphorylates Arpp16 (a splicing variant of Arpp19), on its DSG sequence motif, turning it into a PP2A-B55 inhibitor in the mouse brain (Andrade et al, 2017).

While the *C. elegans* genome does not encode a Gwl kinase, it does encode a single MAST family kinase called KIN-4 (KINase 4) (Figs. 4A and EV3A), mainly expressed in neurons and recently implicated in aging (An et al, 2019) and thermotaxis (Nakano et al, 2022).

We thus hypothesized that KIN-4 might be the long-sought kinase phosphorylating ENSA-1 in worms (Fig. 4B). Consistent with this hypothesis, we found that RNAi-mediated *kin-4* inactivation suppressed the embryonic lethality of the *sur-6*ts allele, similar to ENSA-1 mutation on the DSG motif, which is entirely consistent with KIN-4 acting on ENSA-1 (Fig. 4C). We corroborated these observations obtained using RNAi with the *kin-4(tm1049)* allele, which deletes part of the kinase domain (Fig. EV3B). *sur-6*ts; *kin-4(tm1049)* double mutants were broadly viable at restrictive temperature, in sharp contrast to single *sur-6*ts mutant embryos (Fig. 4D). Furthermore, the *kin-4(tm1049)* deletion mutant, similar to *ensa-1* S61A, suppressed the cell cycle timing defects (Figs. 4E and EV2D) and phenotypes associated with the single *sur-6*ts mutant (Fig. 4F).

These genetic observations indicate that KIN-4 acts as a SUR-6$^{PP2A-B55}$ inhibitor, most likely by directly phosphorylating ENSA-1 at the DSG sequence motif.

## ENSA-1 is phosphorylated at the DSG motif in a KIN-4-dependent manner in vivo

To investigate whether KIN-4 phosphorylates ENSA-1 in vivo, we used phosphoproteomics analysis to compare the phosphoproteome in wild-type and *kin-4Δ* mutant animals. We prepared four biological replicates of total protein extracts from WT and *kin-4Δ* worms. After protein digestion with trypsin, we first compared the total proteomes of both strains by tandem mass spectrometry. With a few noticeable exceptions, we found no massive changes in the proteome of WT versus *kin-4Δ* worms (Fig. EV4A). We then affinity-purified phosphopeptides using Fe-NTA affinity chromatography before identification by tandem mass spectrometry

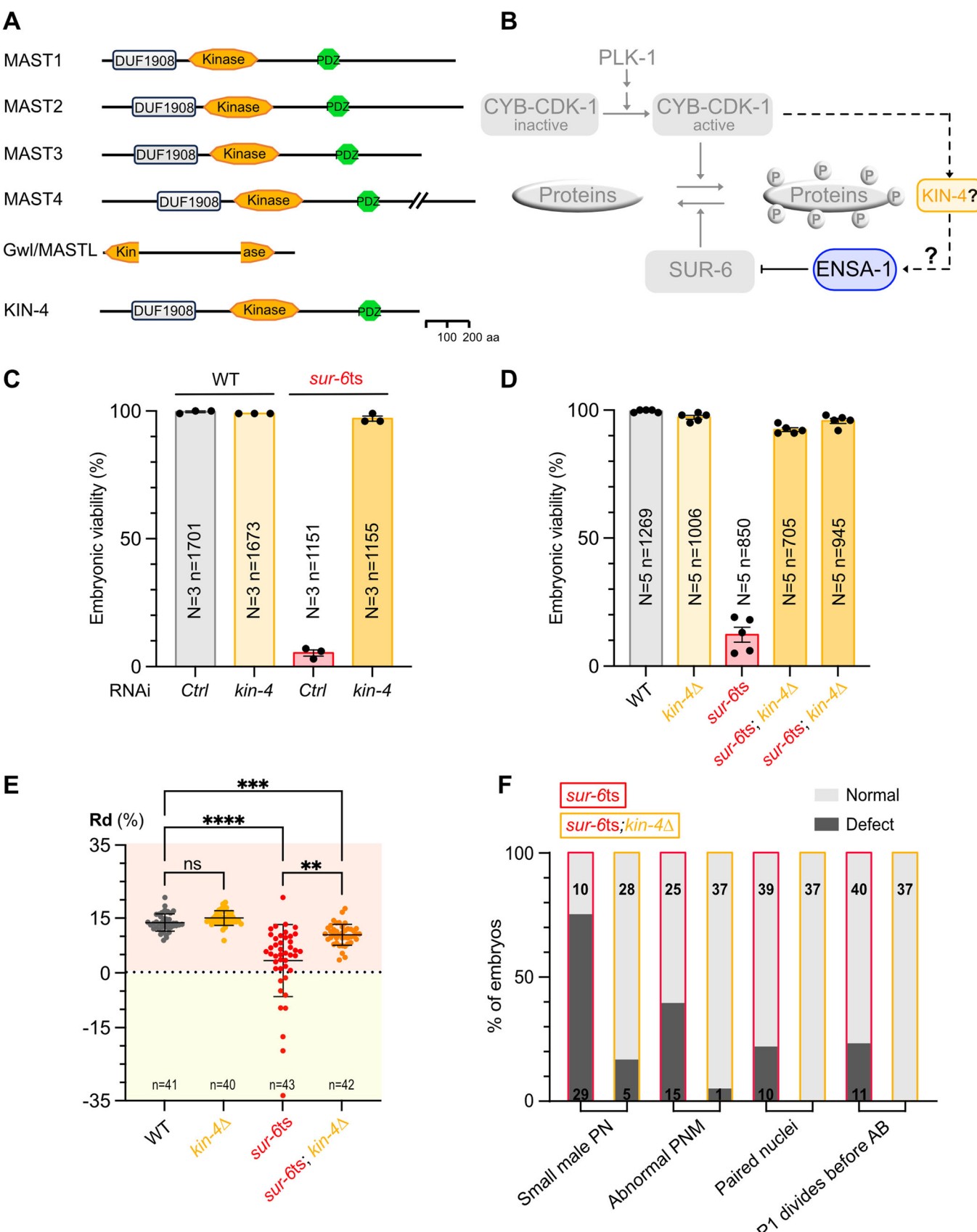

**Figure 4.  *kin-4*, which encodes the single worm MAST kinase, acts as a *sur-6*$^{PP2A-B55}$ inhibitor.**

(A) Schematic representation of MAST kinases domain organization with the domain of unknown function 1908 (DUF1908), the kinase domain (orange), and the postsynaptic density protein-95/disks large/zona occludens-1 (PDZ) domain (green). Mammal genomes encode four MAST kinases plus Gwl/MAST-L, while the *C. elegans* genome encodes only KIN-4. (B) Schematic presenting the potential role of KIN-4 in SUR-6$^{PP2A-B55}$ inhibition. (C) Graphs showing the percentage of embryonic viability, represented as the mean ± standard error to the mean, of wild-type N2 and *sur-6*ts exposed to control or *kin-4* RNAi after shifting the animals 5 h at the restrictive temperature (25 °C). *N* is the number of independent experiments, and *n* is the total number of embryos counted. (D) Graph showing the percentage of embryonic viability, represented as the mean ± standard error to the mean, of wild-type N2, *kin-4Δ (tm1049)*, *sur-6*ts, and *sur-6*ts; *kin-4Δ (tm1049)* double mutants (two clones) cultivated at 16 °C from the L1 stage and shifted 5 h at 25 °C, at adulthood. *N* is the number of independent experiments, and *n* is the total number of embryos counted. (E) Graph presenting the relative difference (Rd) of AB and P$_1$ cell cycle lengths in embryos of the indicated genotypes in percentage, represented as the mean ± standard error to the mean. *n* number of embryos analyzed. Non-parametric tests (Kruskal–Wallis) were used to calculate *p* values, which are displayed as follows: ns = *p* > 0.05; ** = *p* < 0.01; *** = *p* < 0.001; **** = *p* < 0.0001. Exact *p* values from (E) (L-R): *p* < 0.0001, *p* = 0.0003, *p* = 0.0095. Error bars display the standard error to the mean. ns no-significant differences. (F) Graphs presenting the percentage of single *sur-6*ts and double *sur-6*ts; *kin-4Δ (tm1049)* mutant embryos with the indicated phenotypes. Source data are available online for this figure.

(Fig. 5A). The four independent biological replicates allowed quantitative analysis of the enriched phosphopeptides in wild-type (Fig. 5B, gray dots) versus *kin-4* deleted strain (orange dots). We then analyzed the data using a volcano plot and applied a *p* value of 0.01 and a fold change between the two genotypes of 10. Serving as a positive control of the experiment, a KIN-4 phosphopeptide was detected in the wild-type but not in the *kin-4Δ* strain, as expected (Fig. 5B). Phosphorylation of ENSA-1 was systematically detected at the DSG sequence motif in wild-type worms, but not in the *kin-4* null mutant (Fig. 5B).

Beyond identifying ENSA-1 as a KIN-4 substrate in vivo, this quantitative phosphoproteomic analysis revealed additional insights. Notably, a phosphopeptide corresponding to the 6-phosphofructokinase PFK-1.1 was significantly enriched in wild-type worms, suggesting that KIN-4 may phosphorylate PFK-1.1. Interestingly, this phosphosite [RFDS(+79.97)IVPTAGR] is embedded in a sequence presenting similarities to the one surrounding the Serine 61 of ENSA-1 [KFFDS(+79.97)GDYAMDK].

Furthermore, as ENSA-1 and KIN-4 are SUR-6$^{PP2A-B55}$ inhibitors, an increase in SUR-6$^{PP2A-B55}$ activity would be expected in *kin-4* mutants, possibly accompanied by a decrease in phosphorylation of SUR-6$^{PP2A-B55}$ substrate(s) and a reduction in CDK-1 activity, as several positive CDK-1 activators, including CDC-25, are dephosphorylated by PP2A-B55. As an indirect readout of CDK activity, we monitored the level of the inhibitory Y15 phosphorylation on CDK-1. A phosphopeptide covering CDK-1 Y15 was identified by tandem mass spectrometry in WT and *kin-4* mutant worms. Notably, the level of inhibitory Y15 phosphorylation was slightly increased in the *kin-4* mutant, consistent with a reduction of CDK-1 activity in these animals. This reduction of CDK-1 activity does not cause a detectable phenotype in *kin-4Δ* mutant embryos, which are fully viable. However, *kin-4Δ*, *ensa-1 S61A*, and *ensa-1Δ* mutants were slightly more sensitive to partial *cdk-1* depletion by RNAi, although the effect was mild and variable, except for the *ensa-1Δ* mutant (Fig. EV4C), suggesting that CDK-1 activity is indeed reduced upon inactivation of the Gwl-like pathway.

In contrast to CDK-1 Y15, phosphorylation of other proteins, such as SMK-1 or ANI-2, was increased in WT compared to the *kin-4* mutant, consistent with increased SUR-6 activity in the *kin-4* mutant (Fig. 5C). Intriguingly, however, several others proteins specifically accumulated phosphorylated in *kin-4* mutants, indicating that *kin-4* deletion has a broad impact on the phosphoproteome (Fig. 5B). Sequence analysis of the enriched phosphopeptides using

IceLogo (Colaert et al, 2009) revealed that the sites are non-S/T-P sites, they are mainly acidic (Fig. EV4B), suggesting that KIN-4 may regulate directly or indirectly the function of a phosphatase dephosphorylating these sites. This will require further investigation beyond the scope of the present study.

Overall, this quantitative mass spectrometry analysis indicates that ENSA-1 is phosphorylated in vivo at the DSG sequence motif in a KIN-4-dependent manner.

## KIN-4 phosphorylates ENSA-1 at the DSG sequence motif in vitro

Next, we investigated whether KIN-4 phosphorylates ENSA-1 at this motif in vitro. Multiple protein sequence alignments between KIN-4 and MAST1-4 from humans and mice revealed that the kinase domain is well conserved and contains the residues and key features typically found in AGC kinases, as suggested by AlphaFold structural modeling of the kinase domain (Figs. 6A and EV3A) (Hermida et al, 2020). Residues between the DFG motif (light Green Box) and the APE motif (light Green Box) compose the activation loop (Figs. 6A and EV3A). Most eukaryotic kinases are activated by phosphorylation of specific residues in their activation loop (AL). However, it is currently unclear whether MAST kinases require phosphorylation of this region for their activation (Pearce et al, 2010, #91901). Based on sequence analysis, MAST kinases appear to lack a phosphorylatable residue in the activation loop (Somale et al, 2020).

Thus, to determine whether KIN-4 directly phosphorylates ENSA-1 at the DSG sequence motif, we first co-expressed in *E. coli* the KIN-4 kinase domain [amino acids 545-902, hereafter KIN-4$^{Kin-dom}$] and ENSA-1, either wild-type or non-phosphorylatable at the DSG sequence motif (ENSA-1 S61A) (Fig. EV5A). We then probed for ENSA-1 phosphorylation using a commercially available phospho-specific antibody, specifically detecting the phosphorylated DSG motif in Western blot experiments. As shown in Fig. EV5A,B, a specific band was readily detected when KIN-4$^{Kin-dom}$ was co-expressed with ENSA-1 WT (Fig. EV5B lane 1) but not ENSA-1 S61A (lane 4), indicating that KIN-4$^{Kin-dom}$ phosphorylates ENSA-1 at the DSG motif. Mutation of the predicted ATP binding site (K582R) abolished the ability of KIN-4$^{Kin-dom}$ to phosphorylate ENSA-1 (lane 3). Likewise, valine substitution of the Thr-Pro site located in the P+1 loop (KIN-4$^{Kin-dom}$ T730V) abrogated the ability of KIN-4 to phosphorylate ENSA-1 (Fig. EV5B, lane 2).

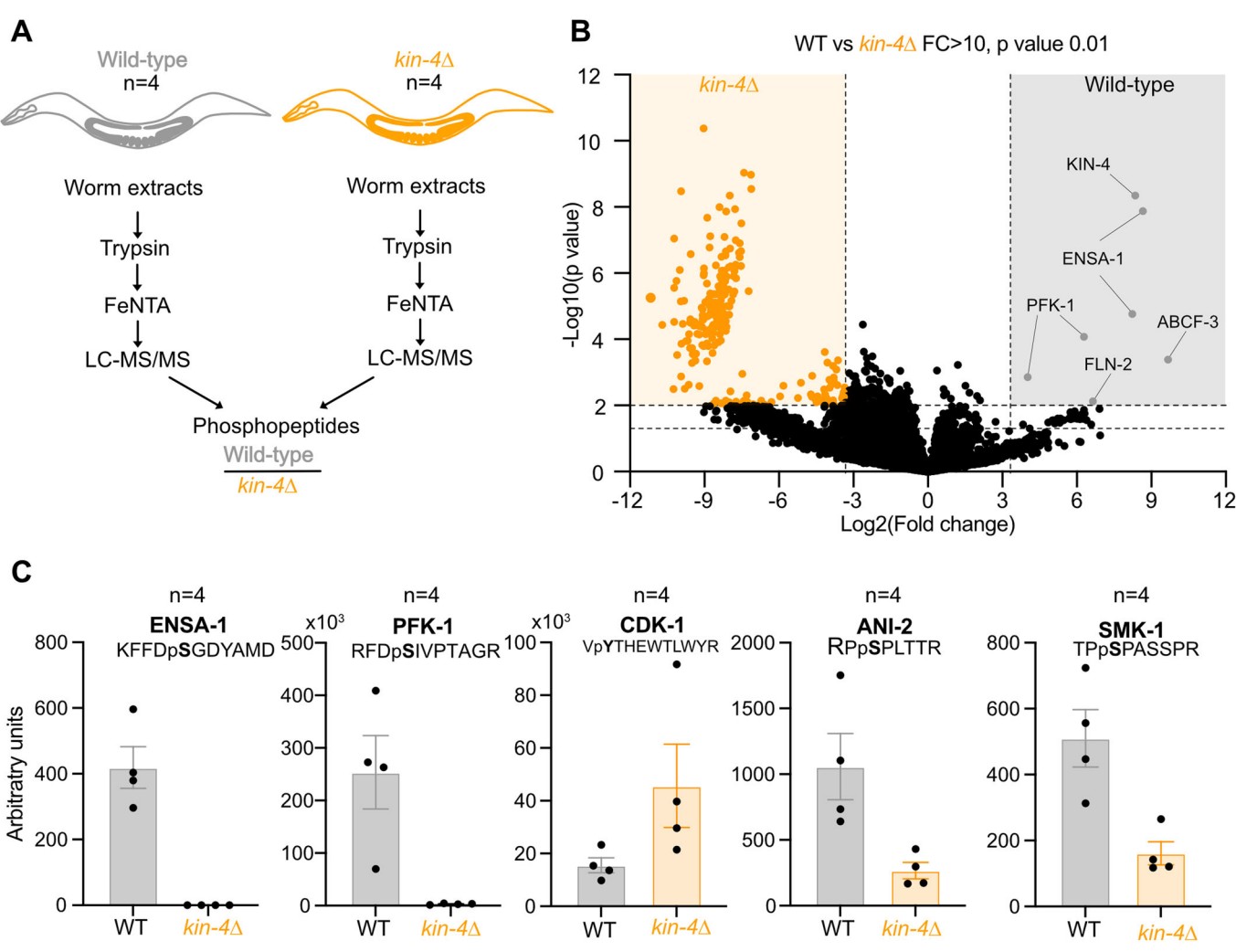

**Figure 5. ENSA-1 is phosphorylated at the DSG motif in a KIN-4-dependent manner in vivo.**

(A) Flowchart of the approach to compare the phosphoproteomes of WT (light gray) versus *kin-4Δ* (orange) animals. Four independent biological replicates ($n = 4$) were analyzed. (B) Visualization of the phosphoproteomic analysis in a Volcano plot. Each point on the graph represents a phosphorylated peptide. The $\log_2$-fold change differences between the WT and *kin-4Δ* were plotted on the x-axis, and the $-\log10$ $p$ value differences were plotted on the y-axis. Phosphopeptides whose abundance is increased in wild-type versus *kin-4Δ* are located to the right of zero on the x-axis, while phosphopeptides whose abundance is decreased are illustrated to the left of zero. Phosphopeptides with statistically significant differential abundance lie above the horizontal threshold ($p = 0.01$). The horizontal dashed lines represent a $p$ value of 0.01 and 0.05 (Student's bilateral $t$-test and assuming equal variance between groups, see also methods section), and the vertical dashed lines show a fold change between WT and *kin-4Δ* of 10. (C) Graphs presenting the abundance (arbitrary value corresponding to the area of the pic detected by LC-MS/MS) of selected phosphopeptides in wild-type versus *kin-4Δ* animals. The phosphopeptide sequence with the position of the phosphosite is indicated at the top of the graph. Error bars display the Standard Error to the mean. $n$ is the number of independent phosphoproteomic analysis. Source data are available online for this figure.

To corroborate these observations, we purified the proteins from *E. coli* and performed in vitro kinase assays. Consistent with the co-expression experiments, KIN-4[Kin-dom] wild-type phosphorylated GST-ENSA-1 but not GST-ENSA-1 S61A, similar to the Xenopus Gwl kinase (Fig. 6B, lane 4 and 8 and Fig. EV5C, lane 4). Mutation of the TP site in the P + 1 loop (T730V) (Fig. 6C, lane 7) or the predicted ATP binding site (K582R) (Fig. 6C, lane 10) abolished the ability of KIN-4[Kin-dom] to phosphorylate ENSA-1.

We confirmed these results by analyzing the kinase reaction by tandem mass spectrometry. ENSA-1 was readily and exclusively phosphorylated at the DSG motif by KIN-4[Kin-dom] in vitro (Fig. EV5D). However, we did not detect phosphorylation on the

KIN-4[Kin-dom], except in the C-terminal extension of the kinase domain, but we could not precisely localize the position of this phosphorylation.

These observations indicate that KIN-4 directly phosphorylates ENSA-1 at the DSG sequence motif in vitro, turning it into a PP2A-B55 inhibitor, and may not require phosphorylation of the activation segment, at least for basal activity (see Discussion).

## KIN-4 can functionally replace Greatwall in the heterologous *Xenopus* system

Although KIN-4 phosphorylates ENSA-1 in worms, it does not contain the atypical activation segment present in Gwl (Fig. 4A).

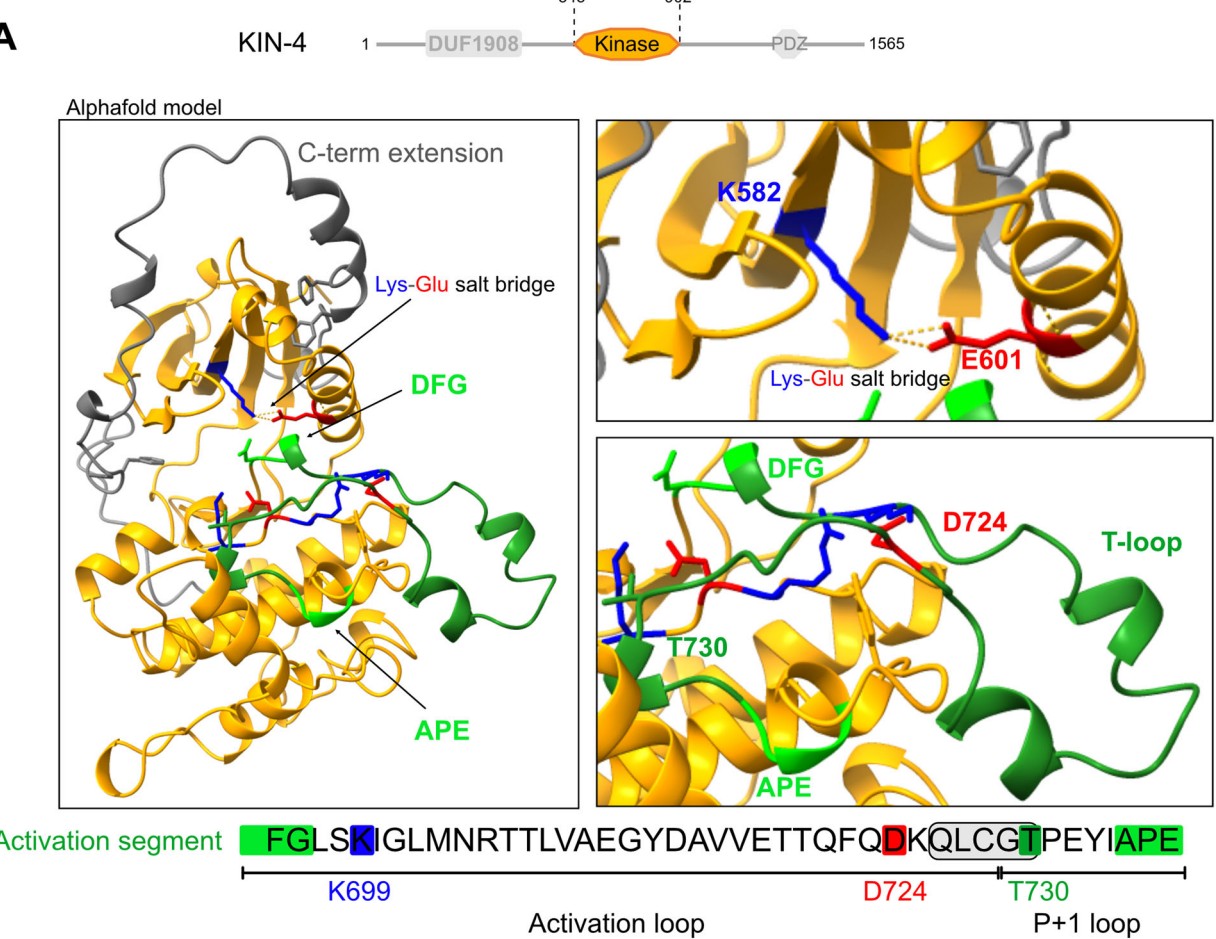

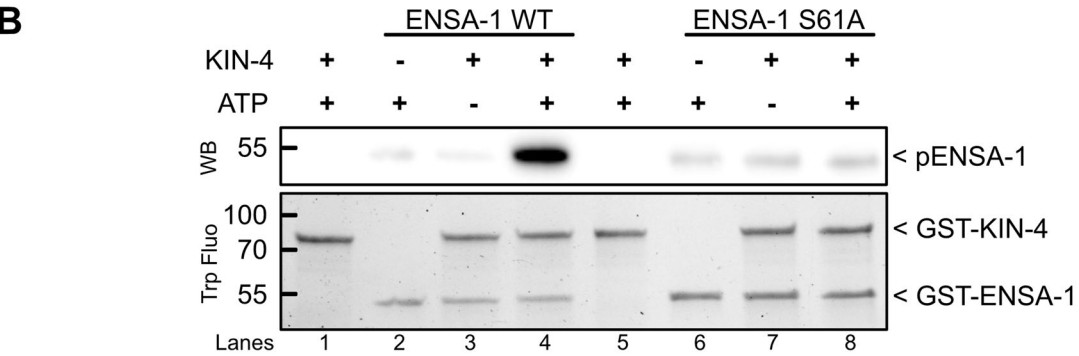

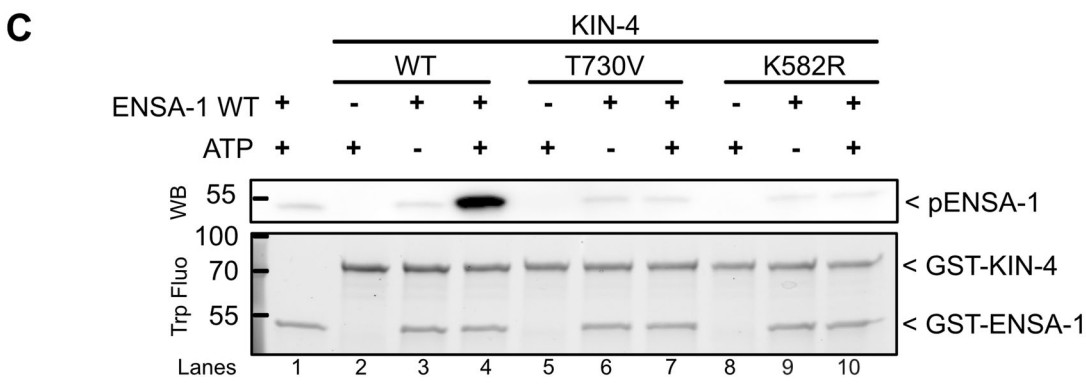

**Figure 6.  KIN-4 phosphorylates ENSA-1 at the DSG motif in vitro.**

(A) AlphaFold structural modeling of the KIN-4 kinase domain. The N and C-lobe of the kinase domain are depicted in orange; the activation segment, flanked by the DFG and APE motifs, is green, and the C-terminal extension is gray. The lysine K582 and the Glutamate E601, which are part of the critical Lys-Glu salt bridge, are indicated. No phosphorylatable residue is present in the activation loop. Instead, the model predicts that the conserved aspartate D724 neutralizes the arginine R675 of the HRD motif and the conserved lysine K699 located just downstream of the DFG motif. In most AGC kinases activated by phosphorylation of the activation loop, this lysine residue directly contacts the phosphorylated residue of the activation loop. (B) Western blot analysis of kinase reactions was carried out with KIN-4$^{Kin-Dom}$ and GST-ENSA-1 WT or S61A as substrates. Blots were probed with antibodies to the phospho-DSG motif. The lower panel shows protein levels detected by tryptophane fluorescence (stain-free, Bio-Rad). (C) Western blot analysis of kinase reactions was carried out with KIN-4$^{Kin-Dom}$ WT, T730V, or K582R and GST-ENSA-1 as substrate. Blots were probed with antibodies to the phospho-DSG motif. The lower panel shows protein levels detected by tryptophane fluorescence (stain-free, Bio-Rad). Source data are available online for this figure.

We thus wondered whether KIN-4 could replace Gwl in the heterologous *Xenopus* system (Fig. 7A, left panel).

Previous work has shown that the addition of a recombinant hyperactive Gwl kinase (Gwl$^{K72M}$) to interphase *Xenopus* egg extracts partially inhibits PP2A-B55, stabilizes substrate phosphorylation by basal Cyclin-Cdk activity, and forces the extract to enter into mitosis (Vigneron et al, 2009; Lorca et al, 2010).

Remarkably, supplementing an interphase egg extract with recombinant worm KIN-4$^{Kin-dom}$ wild-type (Fig. 7A, right panel, compares lanes 9–12 to 1–8) similarly forced the extracts to enter into mitosis, as revealed by the accumulation of phosphorylated (p)Gwl, (p)Cdc25 and (p)Arpp19 proteins, the degradation of Cyclin B2, and the dephosphorylation of Cdk1 on the inhibitory tyrosine 15 (pTyr) (Fig. 7A, lanes 10–12).

This did not occur with the catalytically inactive K582R version (Fig. 7A, compare lanes 9–12 to 1–8). In fact, adding KIN-4$^{Kin-dom}$ to an interphase egg extract immunodepleted of Gwl also induced mitotic entry (Fig. 7B, lanes 5–8), indicating that KIN-4 can functionally replace Gwl in this system.

To corroborate these observations, we asked whether KIN-4 can replace Gwl function during meiosis (Fig. 7C). Xenopus CSF (cytostatic factor arrested) egg extracts prepared from oocytes in the metaphase of meiosis II are typically arrested in a mitotic state with high Cyclin B-Cdk1 (MPF) activity. Gwl kinase is active at this stage and maintains the mitotic state by inhibiting the dephosphorylation of crucial cell cycle regulatory factors by PP2A-B55. However, Gwl immunodepletion releases active PP2A-B55, which can then dephosphorylate Cyclin B-Cdk1 substrates and cause exit from the mitotic state.

As shown in Fig. 7C (lane 2), CSF-arrested extracts exited the mitotic state upon Gwl immunodepletion, as reflected by the dephosphorylated state of Cdc25 and MAP kinase and the accumulation of inactive Cdk phosphorylated on the inhibitory site tyrosine 15 (pTyr). Addition of recombinant KIN-4$^{Kin-dom}$ WT (Fig. 7C, lanes 2–5), but not the catalytically dead KIN-4$^{Kin-dom}$ K583R mutant (lanes 6–9), restored the mitotic state within 20 min. This was accompanied by the accumulation of phosphorylated Cdc25 and MAP kinase, the dephosphorylation of Cdk1 on tyrosine 15, and the phosphorylation of endogenous Arpp19 on S71 by KIN-4 (Fig. 7C). Taken together, these results indicate that *C. elegans* KIN-4$^{Kin-dom}$ can replace Gwl function in Xenopus oocyte extracts. Taken at face value, all these results indicate that KIN-4 is a functional Greatwall homolog.

## KIN-4 and ENSA-1 negatively regulate SUR-6$^{PP2A-B55}$ activity during vulva development in *C. elegans*

The above results showed that by phosphorylating ENSA-1, KIN-4 inhibits SUR-6$^{PP2A-B55}$ activity and regulates cell cycle progression.

However, *sur-6* was initially identified as a positive regulator of RAS signaling during vulva development (Sieburth et al, 1999) (Fig. 8A). We, therefore, asked whether KIN-4 and ENSA-1 were also regulating SUR-6 activity in this developmental context. A gain-of-function mutation in *let-60$^{ras}$(n1046gf)* leads to a multi-vulvae (*muv*) phenotype, where more than three vulval precursor cells (VPC) adopt vulval cell fates (Beitel et al, 1990). *sur-6* mutation suppresses this phenotype (Sieburth et al, 1999) (confirmed in Fig. 8B,C). Strikingly, we found that *ensa-1* and *kin-4* depletion, by releasing SUR-6$^{PP2A-B55}$ activity, significantly increased the *muv* phenotype of the *let-60(n1046gf)* mutant (Fig. 8B,C). ENSA-1 and KIN-4 act via SUR-6 to induce the *muv* phenotype because animals depleted of *ensa-1* and *sur-6* or *kin-4* and *sur-6* phenocopied *sur-6* inactivation (Fig. 8B,C). These observations indicate that KIN-4 and ENSA-1 do indeed regulate SUR-6$^{PP2A-B55}$ activity during vulva development.

In summary, we have found that KIN-4 inhibits SUR-6$^{PP2A-B55}$ activity to regulate various biological processes both in the cell cycle and development in *C. elegans* and may not require upstream phosphorylation for its activation, in contrast to Gwl (Fig. 8D) (see discussion).

## Discussion

We have identified and functionally characterized a pathway inhibiting PP2A-B55 phosphatase activity during *C. elegans* development. In this pathway, KIN-4, the single MAST kinase family member in worms, phosphorylates ENSA-1 at the canonical DSG sequence motif and turns it into an inhibitor of the SUR-6$^{PP2A-B55}$ phosphatase complex. While this pathway is not essential for viability in normal conditions, it is toxic in worms with compromised SUR-6$^{PP2A-B55}$ activity. Hence, it contributes to regulating the tug-of-war between the Cyclin B-Cdk1 kinase and its counteracting PP2A-B55 phosphatase to ensure robust cell cycle asynchrony in two-cell embryos. Remarkably, while KIN-4 does not contain the atypical activation segment found in bona fide Gwl homologs and may not require phosphorylation for activation, it can nevertheless fully replace Gwl's function in Xenopus in promoting mitotic entry in egg extracts and maintaining the mitotic state in CSF-arrested oocytes. Thus, our results unequivocally identify a Greatwall-like pathway in *C. elegans*.

### A balance between kinase and phosphatase activities ensures cell cycle asynchrony during *C. elegans* early embryonic development

While the Cyclin B-Cdk1 kinase and its counteracting PP2A-B55 phosphatase orchestrate mitotic entry, progression, and exit, how these

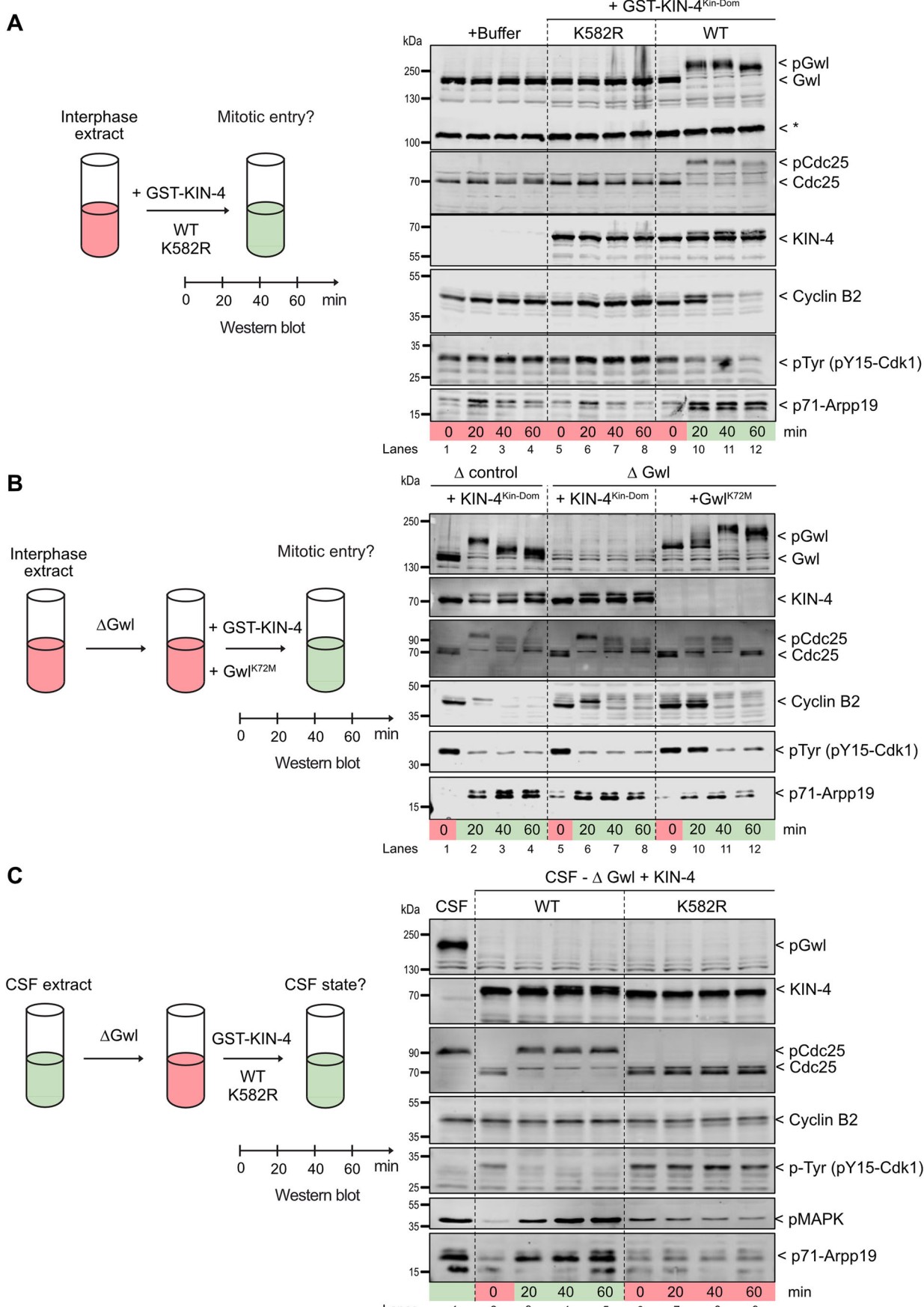

◄ **Figure 7. KIN-4$^{Kin-Dom}$ functionally replaces Gwl in *Xenopus laevis* extracts.**

(A) KIN-4$^{Kin-Dom}$ promotes mitotic entry in interphase egg extracts. Interphase egg extracts were supplemented with either XB Buffer or GST-KIN-4$^{Kin-Dom}$ K582R (K/R) mutant (220 ng/ul final concentration) or GST-KIN-4$^{Kin-Dom}$ WT (at a final concentration of 150 ng/ul). Samples were collected at different time points (0, 20, 40, and 60 min) and analyzed by Western blot using anti-Gwl, anti-Cdc25, anti-Cyclin B2, anti-pTyr-Cdk1, anti-p71 Arpp19 (from top to bottom). Mitotic entry was determined by analyzing the dephosphorylation of Tyr15 of Cdk1-Cyclin B, the phosphorylation of Gwl, Cdc25, and Arpp19 on S71 residue, followed by a subsequent exit of mitosis as indicated by the degradation of Cyclin B. Asterisks denote non-specific band. (B) The KIN-4$^{Kin-Dom}$ protein rescues the phenotype induced by Gwl depletion in interphase. Interphase egg extracts were depleted of Gwl and supplemented with the protein GST-KIN-4$^{Kin-Dom}$ or GST-Gwl K72M, an active form of Gwl. As a control, interphase egg extracts were depleted using purified GST antibodies, and mitotic entry was induced by adding GST-KIN-4$^{Kin-Dom}$. Mitotic entry was determined as described in (A). (C) KIN-4$^{Kin-Dom}$ maintains the mitotic state by promoting PP2A-B55 Inhibition. Depletion of Gwl from mitotic egg extracts (CSF cytostatic factor) induced the loss of the mitotic state. Depleted egg extracts were supplemented with GST-KIN-4$^{Kin-Dom}$ or GST-KIN-4$^{Kin-Dom}$ K/R mutant. The ability of KIN-4$^{Kin-Dom}$ to rescue mitotic exit was assessed by analyzing the stability of cyclin B and the phosphorylation of Cdc25, Tyr15 of Cdk1-cyclin B, P-MAPK, and S71 of Arpp19. Source data are available online for this figure.

competing activities regulate lineage-specific cell cycle duration during animal development remains poorly understood. Previous work showed that Cyclin B-Cdk1 contributes to asynchronous cell division in 2-cell *C. elegans* embryos (Budirahardja and Gonczy, 2008). CDK-1 itself is not asymmetrically localized, but its activators, the phosphatase CDC-25.1 and the Polo-like Kinase PLK-1, are enriched in the AB blastomere (Budirahardja and Gonczy, 2008) such that CDK-1 is activated earlier in the AB than in the P$_1$ blastomere. Moreover, as the AB blastomere displays higher CDK-1 activity than P$_1$, it is less sensitive to partial RNAi-mediated *cdk-1* or *plk-1* depletion than the P$_1$ blastomere. Indeed, in embryos partially depleted of PLK-1 or CDK-1, the division of the P$_1$ blastomere is relatively more delayed than that of the AB blastomere. In the current study, we found that inactivation of the counteracting SUR-6$^{PP2A-B55}$ phosphatase results in the opposite phenotype, with the cell cycle of the AB blastomere comparatively more delayed than P$_1$. PP2A-B55 has been implicated in the dephosphorylation of Cyclin B-Cdk1 activators, such as CDC-25, and other substrates, such as condensins (Yeong et al, 2003). Several hyperphosphorylated substrates may accumulate in the AB blastomere, particularly during mitotic exit, causing the observed cell cycle delay. Consistently, we noticed abnormal elongation of the mitotic spindle with severe defects in chromosome decondensation in *sur-6*ts embryos, consistent with the role of PP2A-B55 function in dephosphorylating targets to promote mitotic exit.

## Identification of a Greatwall-like pathway in *C. elegans*

While the Greatwall pathway has been thoroughly characterized, first in *Drosophila melanogaster*, and *Xenopus laevis*, and then in other systems (Yu et al, 2004; Vigneron et al, 2009; Lorca et al, 2010; Glover, 2012), it has been neglected in *C. elegans*, primarily because *ensa-1* null animals are viable, and there is no obvious Greatwall orthologue in the genome (Kim et al, 2012). The critical role of SUR-6$^{PP2A-B55}$ in cell cycle regulation (Kao et al, 2004; O'Rourke et al, 2011), centrosome maturation (Kitagawa et al, 2011; Song et al, 2011), and other developmental processes, including vulva development (Sieburth et al, 1999; Kao et al, 2004), was, however, well documented, raising questions about the mechanisms regulating PP2A-B55 activity in worms. We exploited the *sur-6*ts allele (O'Rourke et al, 2011) to reveal the existence of a Greatwall-like pathway in worms. At restrictive temperatures, the SUR-6 W140R mutation likely destabilizes the PP2A-B55 complex. Consistent with this prediction, we found using AlphaFold 2 structural modeling that the evolutionarily conserved Tryptophan W140, mutated to Arginine in the *sur-6*ts mutant, is buried in the

complex and located near the interface between SUR-6 and the scaffold PAA-1 subunit (Fig. EV6). Replacing a buried Tryptophan with a positively charged Arginine most likely destabilizes the SUR-6 - PAA-1 interface and possibly the entire SUR-6$^{PP2A-B55}$ complex at the restrictive temperature (25 °C) (Fig. EV6). Still, the phenotypes of *sur-6*ts embryos maintained 48 h at 25 °C are not as severe as the phenotypes of *sur-6* null mutants, indicating that *sur-6*ts embryos retain some residual PP2A-B55 activity, which can be upregulated upon the simultaneous inactivation of *ensa-1* or *kin-4*, demonstrating that these genes act as negative regulators of *sur-6*. Indeed, the cell cycle defects and embryonic lethality of *sur-6*ts mutants were robustly suppressed by the concomitant inactivation of *ensa-1* or *kin-4*, demonstrating that the Greatwall-like pathway controls SUR-6$^{PP2A-B55}$ activity to ensure correct cell division.

Why is the ENSA-1 and KIN-4 pathway dispensable in worms? In Xenopus egg extracts, the first embryonic divisions depend on the oscillations of high Cyclin B-Cdk1 activity to trigger mitotic entry and progression, immediately followed by high PP2A-B55 phosphatase activity required to dephosphorylate Cyclin B-Cdk1 substrates to trigger mitotic exit. During mitosis, the Greatwall pathway neutralizes PP2A-B55 activity downstream of Cyclin B-Cdk1. Then, during mitotic exit, PP1 dephosphorylates Gwl, which relieves the inhibition by pEnsa/pArpp19, and restores the high PP2A-B55 activity required for interphase (Heim et al, 2015; Ma et al, 2016).

In worms, the situation is slightly different. Although Cyclin B-Cdk1 drives mitotic entry and progression (van der Voet et al, 2009; Lara-Gonzalez et al, 2024), it has a less prominent role than in other metazoans. For instance, lamina depolymerization, a crucial step of mitosis, is entirely driven by the Polo kinase PLK-1 in worms and not by Cyclin B-Cdk1 (Velez-Aguilera et al, 2020). Likewise, PLK-1 is the central kinase driving nuclear pore complex disassembly in *C. elegans* embryos (Nkombo Nkoula et al, 2023). Therefore, the levels of PP2A-B55 phosphatase activity required to counteract Cyclin B-Cdk1 are probably lower in worms. Consequently, Cyclin B-Cdk1 does not necessarily require activating the Greatwall-like pathway to reduce PP2A-B55 activity and control cell cycle progression. Nevertheless, this pathway is likely necessary to set a threshold and prevent excessive PP2A-B55 activity that might otherwise be detrimental. Consistent with these ideas, the measured PP2A-B55 phosphatase activity is reduced in worms, to only 1/3 the level found in Xenopus egg extracts (Kim et al, 2012). Furthermore, work in Drosophila has shown that heterozygosity for loss-of-function mutations of PP2A-B55 fully suppresses the effects of Endos or Gwl mutations. Thus, removing one copy of PP2A-B55

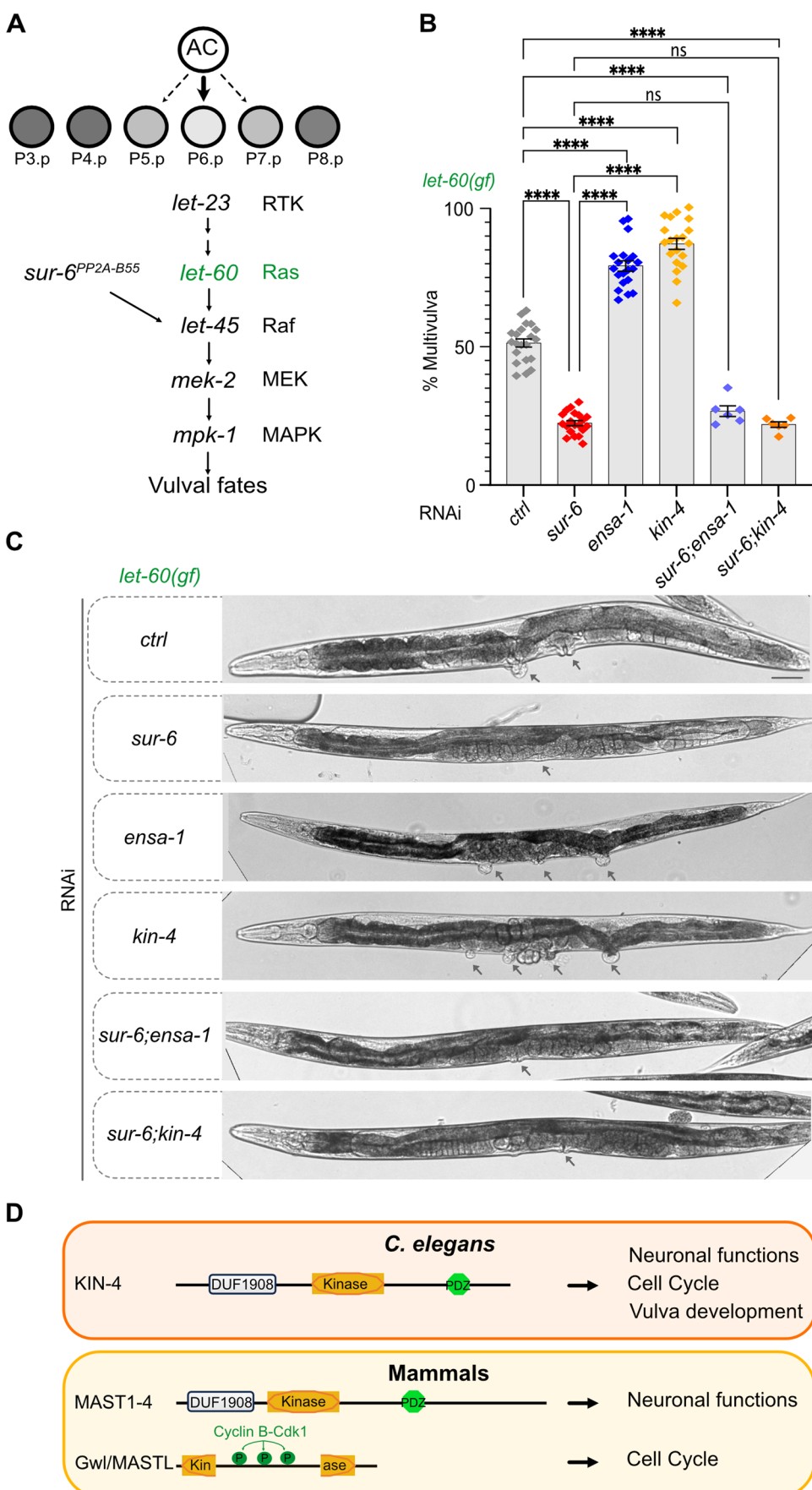

**Figure 8. KIN-4 and ENSA-1 regulate SUR-6^PP2A-B55 activity during cell cycle progression and animal development.**

(A) Schematic of the anchor cell (AC), the vulva precursor cells (VPCs) (top panel), and the RAS pathway regulating *C. elegans* vulva development (bottom panel). Six vulval precursor cells are competent to adopt vulval fates, but only three of them (P5.p, P6.p, and P7.p) do so in response to cell signaling events in wild-type worms. In *let-60^ras*, loss-of-function mutants, fewer than the three VPCs adopt vulval fates, while more than three VPCs adopt the vulval fates in *let-60^ras* gain-of-function mutants. *sur-6^PP2A-B55* is a positive regulator of the *let-60^ras* pathway during vulval development. (B) Graph showing the percentage of *let-60(gf)* worms displaying the multivulva phenotype upon inactivation of *sur-6, ensa-1, kin-4,* or *the* double inactivation *sur-6/ensa-1 and sur-6/kin-4* by RNAi, represented as the mean ± standard error to the mean. The data presented were collected from five independent experiments for the first four conditions and from two experiments for the double RNAi. Ordinary one-way ANOVA multiple comparisons was used to calculate *p* values which are displayed as follows: ns = *p* > 0.05; **** = *p* < 0.0001. Exact *p* values from (B) (L-R); *p* < 0.0001, *p* < 0.0001, *p* < 0.0001, *p* < 0.0001, *p* < 0.0001, *p* < 0.0001, *p* < 0.0001. Error bars display the standard error to the mean. ns no-significant differences. (C) Representative Nomarski images of *let-60(gf)* animals exposed to control, *sur-6, ensa-1, kin-4, sur-6/ensa-1 and sur-6/kin-4* RNAi. The anterior is to the left, the posterior is to the right, and the ventral is at the bottom. Arrows point to the vulvae. Scale bar = 50 μm (D) Model of SUR-6^PP2A-B55 regulation by KIN-4, ENSA-1, and implications for *C. elegans* biology and MAST kinase function in worms and humans. Source data are available online for this figure.

is sufficient in Drosophila to bypass the need for Greatwall (Kim et al, 2012). With respect to PP2A-B55 activity, *C. elegans* could thus be considered equivalent to a fly heterozygous for PP2A-B55. These results have important implications for the evolution of the mechanisms modulating the removal of CDK-mediated phosphorylation. It is only in conditions of high PP2A-B55 activity that the Greatwall pathway is critically required to buffer and counteract it.

## Regulation of KIN-4 and MAST kinase activity

One of the most surprising observations of our study was that recombinant KIN-4^kin-dom, purified from *E. coli* specifically phosphorylated ENSA-1 at the DSG sequence motif, fully rescued the absence of Greatwall in *Xenopus* mitotic and meiotic extracts.

Previous work established that phosphorylation of multiple S/T-P sites by Cyclin B-Cdk1 in the exceptionally large activation segment potentiates Gwl kinase activity (Vigneron et al, 2011; Blake-Hodek et al, 2012; Hermida et al, 2020). Likewise, in 43 out of 61 human AGC kinases, phosphorylation of the consensus sequence (S/T-x-x-G-T) by the 3-Phosphoinositide-dependent protein kinase-1 (PDK1) stimulates kinase activity (Somale et al, 2020). However, no phosphorylatable residues are present in this sequence in MAST kinases (e.g., KIN-4 sequence: Q-V-C-G-T) (Figs. 6A and EV2A). MAST kinases do contain a conserved Thr-Pro site in the P + 1 loop just upstream of the APE motif (Figs. 6A and EV2A). The equivalent site in Gwl is provided by autophosphorylation and is critical to dictate substrate specificity (Hermida et al, 2020). Mutation of this TP site (T730V) abrogates KIN-4 kinase activity. However, we never detected phosphorylation at this site, neither in embryo extracts nor in kinase assays in vitro. Furthermore, the phosphomimetic mutation T730D similarly abrogated kinase activity, leading to inconclusive data regarding the potential role of phosphorylation at this site. Except for this residue, there is no other consensus site for Cyclin B-Cdk1 in the activation loop, and no other phosphosites are predicted in this region (Somale et al, 2020), suggesting that KIN-4 and MAST kinases may not require priming phosphorylation for activation.

## Greatwall-like pathways employing KIN-4 and ENSA-1 are regulating PP2A-B55 in different biological contexts

Beyond regulating cell cycle progression, KIN-4 and ENSA-1 control several other processes in worms. As we show here, KIN-4 and ENSA-1 positively regulate, via SUR-6^PP2A-B55 inhibition, the *ras*

pathway during vulva development. It is noteworthy that *kin-4* has been previously identified in a synthetic lethal screen with the SynMuvB gene *efl-1*, which encodes the E2F4 transcription factor (Lehner et al, 2006). As *efl-1* is also a positive regulator of the RAS pathway during vulva development, simultaneous inactivation of both genes may cause *ras* pathway hyperactivation and synthetic lethality. KIN-4 is also abundant in neurons in *C. elegans*, where it controls lifespan and thermotaxis (An et al, 2019; Nakano et al, 2022) and recent work identified a role of ENSA-1 in chemotaxis (Tomioka et al, 2023). It seems likely that KIN-4 and ENSA-1 regulate these processes by inhibiting SUR-6^PP2A-B55 phosphatase activity, though proof awaits further investigation. As the single MAST kinase of the *C. elegans* genome is not essential for animal survival, it represents an attractive model system to dissect the role and regulation of MAST kinases in a developmental context.

## Methods

### Reagents and tools table

| Reagent/resource | Reference or source | Identifier or catalog number |
|---|---|---|
| **Experimental models** | | |
| Xenopus Laevis | Centre de Ressources Biologiques Xenopes-Rennes | http://www.celphedia.eu/en/centers/crb |
| *C. elegans* strains | | Table EV2 |
| **Recombinant DNA** | | |
| PLP2610/pETM30-2 | 6xHis -GST-TEV -KIN-4WT (kinase domain 545-902 /isoform a) | OLP 2923-2924 |
| PLP2728/pETM30-2 | 6xHis -GST-TEV -KIN-4 T730V (kinase domain aa 545-902 / isoform a) | OLP 3124-3125 |
| PLP2729 | 6xHis -GST-TEV -KIN-4 K582R (kinase domain aa 545-902 / isoform a) | OLP 3126-3127 |
| PLP2619/pDest 15 | GST - ENSA-1 WT | Gateway |
| PLP2620/pDest 15 | GST - ENSA-1 S61A | Gateway |
| PLP2651/pCDFDuet-1 | MBP - ENSA-1 /EMPTY | OLP 2997-2998 |
| PLP2701/pCDFDuet-1 | MBP - ENSA S61A /EMPTY | OLP 2929-2930 |
| PLP2565/L4440 | KIN-4 183-349 isoform a | OLP 2863-2864 |
| N/A | L4440 (RNAi Feeding vector) | (Kamath et al, 2003) |
| Arhinger Library | *ensa-1* cloned into L4440 | (Kamath et al, 2003) |
| pLP2565 | *kin-4* cloned into L4440 | This study |
| Arhinger Library | *sur-6* cloned into L4440 | (Kamath et al, 2003) |
| Arhinger Library | *cdk-1* cloned into L4440 | (Kamath et al, 2003) |

| Reagent/resource | Reference or source | Identifier or catalog number |
|---|---|---|
| pLP2984 | *kin-4+sur-6* cloned into L4440 | This study |
| pLP2985 | *ensa-1+sur-6* cloned into L4440 | This study |
| **Antibodies** | | |
| Anti-pENSA-1 | Cell Signaling | 5240S |
| Anti-GST | Cell Signaling | #2622 |
| Anti-MBP | NEB Biolabs | E8032S |
| Peroxidase Goat Anti-Mouse IgG (H + L) | Sigma | Cat#A9917 |
| Peroxidase Goat Anti-Rabbit IgG (H + L) | Sigma | Cat#A0545 |
| Recombinant GST-Human Greatwall K72M mutant | (Vigneron et al, 2011) | N/A |
| Rabbit Polyclonal anti-xenopus Greatwall | (Vigneron et al, 2011) | N/A |
| Monoclonal p44/42 Mapk | Cell Signaling | #9106S |
| Rabbit Polyclonal anti-Xenopus Cdc25 | (Lorca et al, 2010) | N/A |
| Dynabeads protein G | Life Technology | Cat#10004D |
| HRP-conjugated anti-Mouse secondary antibodies | Bio-Rad | Cat#1172-1011 |
| Rabbit Polyclonal Phospho-Cdc2 (Tyr15) | Cell Signaling | Cat#9111 |
| HRP-conjugated anti-Rabbit secondary antibodies | Cell Signaling | Cat#7074 |
| Rabbit Polyclonal anti-Xenopus Cyclin B2 | (Abrieu et al, 1997) | N/A |
| Rabbit Polyclonal anti-GST | (Vigneron et al, 2009) | N/A |
| **Oligonucleotides and other sequence-based reagents** | | |
| OLP 2923-2924 | GGCGCCATGGATAATAGAGCGCCGTGCGAGGATG | ATGCCTCGAGACCTAATCAATGGAAACAATCGAAT |
| OLP 3124-3125 | CAATTGTGTGGAGTTCCAGAGTACATTGCACCTGA | TCAGGTGCAATGTACTCTGGAACTCCACACAATTG |
| OLP 3126-3127 | CGCCAACGTTTTGCTTTGCGAAAGATGAATAAACAG | CTGTTTATTCATCTTTCGCAAAGCAAAACGTTGGCG |
| OLP 2929-2930 | CCAGAAATTTTTCGACGCTGGAGATTATGCAATGG | CCATTGCATAATCTCCAGCGTCGAAAAATTTCTGG |
| OLP 2997-2998 | TATACATATGATGAAAATCGAAGAAGGTAAACTGG | CAGATCTCGAGTTATTCGCTTGTTGCGTTTGGAG |
| OLP 2929-2930 | CCAGAAATTTTTCGACGCTGGAGATTATGCAATGG | CCATTGCATAATCTCCAGCGTCGAAAAATTTCTGG |
| OLP 2863-2864 | AATTCCATGGATGGCTCCTGTCCAAGATACTCGTC | GATTAAGCTTCCACTGCTACTTCCTGATTGTCTTG |
| **Chemicals, Enzymes and other reagents** | | |
| Coomassie R250 | Sigma | Cat#B014925G |
| Ponceau Red | Sigma | Cat#A1405 |
| Amersham™ Protran® Western blotting membranes, nitrocellulose | Merk | GE10600002 |
| Protease inhibitor cocktail | Roche | 74506500 |
| Bio-Rad protein assay | Bio-Rad | #5000001 |

| Reagent/resource | Reference or source | Identifier or catalog number |
|---|---|---|
| IPTG | Euromedex | Cat#EU0008-B |
| Adenosine triphosphate (ATP) | Sigma | Cat#A2383 |
| Glutathione | Sigma | Cat#G4251 |
| PVDF transfer Membrane | Millipore | Cat#88518 |
| Bovine Serum Albumin | Sigma | A2153 |
| 4–20% Mini-PROTEAN® TGX™ Precast Protein Gels | Bio-Rad | #4561096 |
| GSTrap 4B Columns | Cytiva | 28401747 |
| ECL reagent | Millipore | Cat#WBKLS0500 |
| BP Clonase II Enzyme Mix (Gateway cloning) | Invitrogen | Cat#11789-020 |
| LR Clonase II Enzyme Mix (Gateway cloning) | Invitrogen | Cat#11791-020 |
| Pfu | Promega | Cat#M7741 |
| DpnI | Biolabs | Cat#R0176S |
| Alt-R® CRISPR-Cas9 tracrRNA, 100nmol | IDT | Cat#1072534 |
| **Software** | | |
| Clustal W2 | (Larkin et al, 2007) | N/A |
| Jalview | (Waterhouse et al, 2009) | N/A |
| ImageJ | (Schneider et al, 2012) | N/A |
| ZEN | Zeiss | N/A |
| PRISM | Graphpad | N/A |
| Metamorph | Molecular Devices | N/A |
| Affinity Designer 2 | Affinity | N/A |
| Chimera X | RBVI | (Meng et al, 2023) |
| IceLogo | (Colaert et al, 2009) | N/A |
| **Other** | | |
| Spinning disk confocal head CSU-X1 | Yokogawa – Gataca | N/A |
| cMOS prime 95 Camera | Photometrics | N/A |
| Chemidoc™ Gel Imaging system | Bio-Rad | N/A |
| Microscope Axio Observer Z1 | Zeiss | N/A |

## Molecular biology

All the DNA constructs were generated by Gateway cloning or with restriction enzymes. All DNA constructs were verified by sequencing. Oligonucleotides used for site-directed mutagenesis were purchased from Eurofins or Merck Millipore. crRNA and ssODN used for CRISPR/Cas9 were obtained from IDT.

## Worm strains, RNAi, and phenotypic analysis

*C. elegans* strains were cultured and maintained using standard procedures (Brenner, 1974; Stiernagle, 2006). RNAi was performed by the feeding method using HT115 bacteria essentially as

described (Kamath et al, 2001), except that 2 mM of IPTG was added to the NGM plates and in the bacterial culture just before seeding the bacteria. HT115 bacteria were transformed with either empty plasmid (mock RNAi, control) or plasmids expressing *ensa-1, kin-4,* or *sur-6* under the control of the T7 polymerase. RNAi clones were obtained from the Arhinger library (Open Source, BioScience) or were constructed.

### Progeny test

L1 larvae N2, *sur-6*ts, or *sur-6(sv30)* were placed on RNAi plates for 96 h at 16 °C (until young adults) and then shifted at 25 °C for 5 h. Viability assays were performed as follows: ten worms were placed in NGM plated with OP50 bacteria for 3 h and then removed. Eggs were counted and recounted after 24 h. The percentage of viability was then determined as the fraction of embryos hatched among the total progeny.

### muv phenotype analysis

To score the effect of *ensa-1, kin-4* depletion on *ras (let-60(gf))* multivulva phenotype, around 200 synchronized larvae were placed on a single feeding plate and incubated 66–72 h at 20 °C until control worms reached adulthood. Phenotypes were scored when the first eggs laid by control worms started to hatch to ensure that all worms fully developed to the adult stage. Vulval defect phenotypes were scored under a dissecting scope by counting the total number of worms and the number of worms exhibiting multiple vulvae. The counting was repeated once for each plate to minimize scoring errors.

### Microscopy

For cell cycle timing analysis in live specimens by differential interference contrast (DIC) microscopy, embryos were obtained by cutting open gravid hermaphrodites using two 21-gauge needles. Embryos were handled individually and mounted on a coverslip in 3 μl of M9 buffer. The coverslip was placed on a 3% agarose pad. DIC images were acquired by an Axiocam Hamamatsu ICc 1 camera (Hamamatsu Photonics, Bridgewater, NJ) mounted on a Zeiss AxioImager A1 microscope equipped with a Plan Neofluar 100×/1.3 NA objective (Carl Zeiss AG, Jena, Germany), and the acquisition system was controlled by Axiovision software (Carl Zeiss AG, Jena, Germany). Images were acquired at 5-s intervals.

Representative images of the multivulvae were made by transferring worms from a feeding plate onto a drop of 10 mM sodium azide in M9 buffer. After 5 min, immobilized worms were placed on a 3% agarose mounted in between a glass slide and a cover glass. Images were acquired by phase-contrast on an EVOS™ microscope (Invitrogen™) using a Fluorite Plan 10x objective, long working distance of 0.25 NA.

Spinning disk confocal microscopy was performed at 23 °C using a spinning disc confocal head (CSU-X1; Yokogawa Corporation of America) mounted on an Axio Observer. Z1 inverted microscope (Zeiss) equipped with 491- and 561-nm lasers (OXXIUS 488 nm 150 mW, OXXIUS Laser 561 nm 150 mW) and sCMOS PRIME 95 camera (Photometrics). Acquisition parameters were controlled by MetaMorph software (Molecular Devices). In all cases, a 63×, Plan-Apochromat 63×/1.4 Oil (Zeiss) lens was used,

and images were acquired at 10-s intervals. Captured images were processed using ImageJ and Affinity Designer.

### Quantification of cell cycle length

To quantify cell cycle length, embryos of the different genotypes were all recorded on the same day on two microscopes. On average, several independent experiments were performed to record almost 40 embryos for each genotype. We quantified the elapsed time between furrow initiation in $P_0$, AB, and $P_1$ blastomere. Furrow initiation corresponds to the point where the plasma membrane starts to invaginate. During these experiments, the room temperature varied between 21 and 25 °C, resulting in some variability in the quantification of cell cycle lengths.

### CRISPR/Cas9 genome engineering

Genome editing by CRISPR/Cas9 was performed essentially as described (Paix et al, 2015; Dokshin et al, 2018) by injecting in the *C. elegans* gonad a mix of Cas9 protein (0.25 mg/ml), duplex crRNA/tracrRNA (3pmol/ml), ssODN (110 ng/ml), and pRF4 plasmid (40 ng/ml) and screening the edits by PCR.

### ensa-1(bab399)

crRNA: ttccagAAATTTTTCGACTC
ssODN:  TGCGGTTTGGATCCCAATCCTGTTCCAGCTTTCGA CTTGTCCATTGCgTAgTCaCCgGcGTCGAAAAATTTctggaaaata atcgatttcattctgttgtttttaaatatgactaattgt.

### sur-6(lea1)ts

crRNA: ACGAGAAAATTAATCAAATT
  ssODN:GATTATTTGAAATCACTAGAAATTGACGA- GAAAATTAATCAAATTCGTcGACTTAAAAAGAAGAATG- CAGCCAATTTCATTCTCTCGACCAACGAC

### Biochemistry

#### Protein expression and purification

GST-KIN-4[Kin-dom], GST-ENSA-1 full-length, and their variants were purified in the same conditions. Protein expression was induced by the addition of 1 mM of isopropyl β-D-thiogalactopyranoside (IPTG) to 1 L cultures of exponentially growing *E. coli* BL21 DE3 strain (OD = 0.6) before incubation overnight at 18 °C for GST-KIN-4[Kin-dom], or 24 °C for 4 h for GST-ENSA-1.

After pelleting by centrifugation, the bacteria were resuspended in in lysis buffer (25 mM Tris-HCl, pH 8.0, 250 mM NaCl, 5 mM $MgCl_2$, 1 mM TCEP) supplemented with protease inhibitor cocktail (Roche), and broken by sonication. The supernatant was loaded onto a GSTrap 4B 5 ml (Cytiva), and after washing, the protein was eluted using 20 mM Glutathion pH = 8. Before storage, the buffer was exchanged to 25 mM Tris-HCl, pH 7.5, 250 mM NaCl, 5 mM $MgCl_2$, and 0.5 mM TCEP using a G25 column, and the proteins were aliquoted, flash-frozen in liquid nitrogen and stored at −80 °C. The concentration of all the purified recombinant proteins was determined using Bradford quantification.

### Kinase assays

In vitro kinase assays were performed at 30 °C for 30 min in 25 μl containing 25 mM HEPES pH = 7.5, 150 mM NaCl, 10 mM MgCl$_2$, 1 mM DTT, 2.5 μM of GST-KIN-4[Kin-dom] 2.5 μM of GST-ENSA-1, and 1 mM ATP. Reactions were stopped by adding 5 μl of Laemmli buffer before boiling for 5 min at 95 °C. All the samples were loaded on 4–20% SDS-PAGE (TGX Bio-Rad) and run in 1X Tris/glycine/SDS buffer. Proteins were detected using Stain-Free technology and the ChemiDoc MP Imaging System (Bio-Rad) for total gel protein.

### Western blot analyses

Samples were electrophoresed on SDS-PAGE(TGX Bio-Rad) and electro-transferred to nitrocellulose membrane. Next, western blots were probed with primary and appropriate secondary antibodies. All shown western blots are representative of at least three different experiments. Western blots were realized following standard procedures. Antibodies used in this study are listed in the resource table. HRP-conjugated anti-rabbit and anti-mouse antibodies (Sigma-Aldrich) were used at 1:3000, and the signal was detected by chemiluminescence (Millipore) with ChemiDoc MP Imaging System (Bio-Rad).

## Preparation of *Xenopus* egg extract

Frogs were obtained from « TEFOR Paris-Saclay, CNRS UMS2010 / INRAE UMS1451, Université Paris-Saclay», France, and kept in a *Xenopus* research facility at the CRBM (Facility Centre approved by the French Government. Approval n° C3417239). Females were injected with 500 U Chorulon (Human Chorionic Gonadotrophin), and 18 h later, laying oocytes were used for experiments. Adult females were exclusively used to obtain eggs. All procedures were approved by the Direction Generale de la Recherche et Innovation, Ministère de L'Enseignement Supérieur de la l'Innovation of France (Approval n° APAFIS#40182-202301031124273v4).

Mitotic egg extracts (CSF) were prepared from unfertilized Xenopus eggs arrested at the metaphase of the second meiotic division, as described (Lorca et al, 1991).

To prepare Interphase egg extracts, dejellied unfertilized eggs were transferred in XB buffer (100 mM KCl, 0.1 mM CaCl$_2$, 1 mM MgCl$_2$, 50 mM Sucrose, 10 mM HEPES, pH 7.7). Eggs were activated using a calcium ionophore (Sigma A23187: 2 ug/ml final concentration) and crushed 30 min later by centrifugation (10,000 × g 30 min). The cytoplasmic fraction was recovered with a needle and centrifuged (10,000 × g 30 min). The clear cytoplasmic fraction was aliquoted and stored at −70 °C.

All the experiments were performed with 23 μl of Xenopus extracts supplemented with 3 μl of KIN-4 WT [1.43 μg/ul], 3 μl KIN-4 K582R [3.48 μg/ul], or 3 μl of hyperactive Gwl K72M [140 ng/ul].

## Immunoprecipitation

Immunodepletions were performed using 25 μl of extracts, 20 μl of magnetic Protein G-Dynabeads (Dynal), and 2 μg of each antibody. Antibody-linked beads were washed twice with XB buffer and incubated for 20 min at RT with 25 μl Xenopus egg extracts. For immunodepletion, the supernatant was recovered and used for subsequent experiments.

## Worm cultures and extracts

Liquid cultures of N2 or kin4D worms synchronized at the L1 stage were grown on an S medium using E. coli HB101 bacteria as a food source. Worms were harvested at the young adult stage by filtration using nylon mesh (Sefar Nitex, 35-mm mesh). After several washing steps with M9 buffer, the worms were resuspended in one volume of 100 mM NaCl, frozen as beads in liquid nitrogen following cryo-lysis by cryogenic grinding (RETSCH MM400), and kept at −80 °C essentially as described (Velez-Aguilera et al, 2024). Four cryo-lysates of each strain were prepared from four independent liquid cultures. 200 mg of each worm cryo-lysed was resuspended in 400 μl of lysis buffer [37 mM Tris-HCl pH 7.5, 100 mM NaCl, 3 mM MgCl$_2$, 1.5 mM DTT, phosphatase inhibitors (PhosSTOP EASYpack, Roche), and protease inhibitors (cOmplete Protease Inhibitor Cocktail tablets, Roche)] and incubated with 640 U of benzonase nuclease (Sigma-Aldrich) for 30 min at 4 °C. The lysates were clarified by centrifugation (twice 13,000 × g, 20 min at 4 C°), and protein concentration was measured.

## Phosphopeptide enrichment prior to LC-MS/MS analysis

About 250 μg of proteins from each worm extract were precipitated using six volumes of cold acetone (−20 °C). Vortexed tubes were incubated overnight at −20 °C then centrifuged for 10 min at 11,000 rpm, 4 °C. The protein pellets were dissolved in 8 M urea + 25 mM NH$_4$HCO$_3$ buffer pH 7.8. Samples were then reduced with DTT 10 mM final and alkylated with 20 mM MMTS, then diluted below 1 M urea before trypsin digestion overnight at 37 °C (enzyme/substrate ratio of 1/50). Beads from pulldown experiments were incubated overnight at 37 °C with 100 μL of 25 mM NH$_4$HCO$_3$ buffer containing 4 μg of sequencing-grade trypsin.

The digested peptides were desalted on Sep-Pak classic C18 cartridges. Cartridges were sequentially washed with 2 mL methanol, 2 mL of a 70% (v/v) aqueous ACN containing 1% (v/v) TFA and equilibrated with 2 mL of 1% (v/v) aqueous TFA. Peptides were acidified with 1% (v/v) aqueous TFA, applied to the column, and washed with 2 × 1 mL of a 1% (v/v) aqueous FA. Peptides were eluted by applying 2 × 500 μL of a 70% (v/v) aqueous ACN containing 0.1% (v/v) FA. Eluates were vacuum-dried. Phospho-peptide enrichment was performed according to the manufacturer's procedure, starting from the lyophilized peptide samples. Eluted phosphopeptides and peptides from the load column flow-through were loaded and desalted on evotips provided by Evosep (Odense, Denmark) according to the manufacturer's procedure before LC-MS/MS analysis.

## Mass spectrometry analysis of in vitro kinase assays

Samples were diluted in urea 8 M - NH$_4$HCO$_3$ 25 mM buffer, then reduced with DTT 10 mM final and alkylated with MMTS 20 mM final. After a 16-fold dilution in NH$_4$HCO$_3$, samples were digested overnight at 37 °C by a mixture of trypsin/Lys C (1/20 Enzyme/Substrate ratio). The digested peptides were loaded and desalted on evotips provided by Evosep one (Odense, Denmark) according to the manufacturer's procedure.

## LC-MS/MS acquisition

Samples were analyzed on a timsTOF Pro 2 mass spectrometer (Bruker Daltonics, Bremen, Germany) coupled to an Evosep one system (Evosep, Odense, Denmark) operating with the 30SPD method developed by the manufacturer. Briefly, the method is based on a 44-min gradient and a total cycle time of 48 min with a C18 analytical column ($0.15 \times 150$ mm, 1.9 µm beads, ref EV-1106) equilibrated at 40 °C and operated at a flow rate of 500 nL/min. $H_2O$/0.1% FA was used as solvent A, and ACN/ 0.1% FA was used as solvent B.

The timsTOF Pro 2 was operated in PASEF mode[1] over a 1.3 s cycle time. Mass spectra for MS and MS/MS scans were recorded between 100 and 1700 $m/z$. Ion mobility was set to 0.75–1.25 V·s/cm$^2$ over a ramp time of 180 ms. Data-dependent acquisition was performed using 6 PASEF MS/MS scans per cycle with a near 100% duty cycle. Low $m/z$ and singly charged ions were excluded from PASEF precursor selection by applying a filter in the $m/z$ and ion mobility space. The dynamic exclusion was activated and set to 0.8 min, a target value of 16,000 was specified with an intensity threshold of 1000. Collisional energy was ramped stepwise as a function of ion mobility.

## Data analysis

MS raw files were processed using PEAKS Online 11 (build 1.9, Bioinformatics Solutions Inc.). Data were searched against the Caenorhabditis Elegans Wormpep release 2021_07 database (total entry 28411). Parent mass tolerance was set to 20 ppm, with fragment mass tolerance at 0.05 Da. Specific tryptic cleavage was selected, and a maximum of two missed cleavages was authorized. For identification, the following post-translational modifications were included: acetyl (Protein N-term), oxidation (M), deamidation (NQ) and phosphorylation (STY) as variables and Beta-methylthiolation (C) as fixed. Identifications were filtered based on a 1% FDR (false discovery rate) threshold at both peptide and protein group levels. Label-free quantification was performed using the PEAKS Online 11 quantification module, allowing a mass error tolerance of 10 ppm, a CCS error tolerance of 0.02, and 0.5 min of retention time shift tolerance for the match between runs. Protein abundance was inferred using the top N peptide method, and TIC was used for normalization. Multivariate statistics on protein or peptide measurements were performed using Qlucore Omics Explorer 3.9 (Qlucore AB, Lund, SWEDEN). A positive threshold value of 1 was specified to enable a log2 transformation of abundance data for normalization, i.e., all abundance data values below the threshold will be replaced by 1 before transformation. The transformed data were finally used for statistical analysis, i.e., evaluation of differentially present proteins or peptides between two groups using a Student's bilateral $t$-test and assuming equal variance between groups. A $p$ value better than 0.01 was used to filter differential candidates (Fig. 5B).

## Protein structure modeling

C. elegans PP2A-B55 complex was produced using Alphafold Multimer version 2.2 (preprint. EVANS et al, 2022).

## Reproducibility, quantification, and statistical analysis

All experiments presented in this manuscript have been repeated at least three times except the LC-MS/MS analysis of WT versus kin-4Δ worms performed once, using four independent extracts from different worm cultures.

Statistical analyses were performed using the Prism software (GraphPad).

## Data availability

The data were available in the primary article. The corresponding author can provide original data, C. elegans strains, and plasmids generated in this study upon request.

The mass spectrometry proteomics data have been deposited to the ProteomeXchange Consortium via the PRIDE (http://www.ebi.ac.uk/pride) (Perez-Riverol et al, 2022) partner repository with the dataset identifier PXD054396.

The source data of this paper are collected in the following database record: biostudies:S-SCDT-10_1038-S44318-025-00364-w.

## Peer review information

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

## Acknowledgements

We thank P. Moussounda for helping with media preparation. R. Karess and M. Peter for critical reading of the manuscript, and W. Grange for the statistical analysis. We also thank A. Pillan and E. Beaumale for stimulating discussions and advice. We acknowledge the ImagoSeine core facility of the Institut Jacques Monod, member of IBiSA and France-BioImaging (ANR-10-INBS-04) infrastructures, and the Institut Jacques Monod's "Structural and functional proteomic platform." This work was supported by the French Infrastructure for Integrated Structural Biology (FRISBI) ANR-10-INBS-0005. Some strains were provided by the Caenorhabditis Genetics Center (CGC), funded by the NIH Office of Research Infrastructure Programs (P40 OD010440). Some strains were generated by SEGiCel (SFR Santé Lyon Est CNRS UAR 3453, Lyon, France) with the support of CNRS and IBiSA. We also acknowledge the help of the ZEFIX animal facility. NJ is supported by a grant from the Fondation ARC pour la Recherche sur le Cancer (ARCPJA2022050005002) and a grant from Université Paris-Cité IDEx (ANR-18-IDEX-0001, "Emergence en Recherche" RS30J23IDX64_KATAREP). Work in the laboratory of TL is supported by grants from "Agence Nationale pour la Recherche" (ANR, France - ANR-22-CE13-0022) and by a grant from the Fondation ARC pour la Recherche sur le Cancer (PJA3 TL ARCPJA2023060006673). Work in the laboratory of LP is supported by grants from "Agence Nationale pour la Recherche" (ANR, France - ANR-22-CE13-0022) and by the "Ligue Nationale Contre le Cancer" (Equipe labéllisée, France).

## Author contributions

**Ludivine Roumbo**: Conceptualization; Resources; Data curation; Formal analysis; Validation; Investigation; Visualization; Methodology; Writing—review and editing. **Batool Ossareh-Nazari**: Conceptualization; Formal analysis; Supervision; Validation; Investigation; Methodology; Project administration; Writing—review and editing. **Suzanne Vigneron**: Formal analysis; Investigation. **Ioanna Stefani**: Investigation; Methodology. **Lucie Van Hove**: Resources; Investigation; Methodology. **Véronique Legros**: Formal analysis; Validation; Investigation; Methodology. **Guillaume Chevreux**: Formal analysis; Validation; Investigation; Methodology. **Benjamin Lacroix**: Formal analysis; Investigation; Methodology. **Anna Castro**: Investigation. **Nicolas Joly**: Conceptualization; Formal analysis; Supervision; Methodology; Writing—review and editing. **Thierry Lorca**: Conceptualization; Resources; Formal analysis; Supervision; Funding acquisition; Validation; Visualization; Methodology; Project administration; Writing—review and editing. **Lionel Pintard**: Conceptualization; Data curation; Formal analysis; Supervision; Funding acquisition; Validation; Investigation; Visualization; Methodology; Writing—original draft; Project administration; Writing—review and editing.

Source data underlying figure panels in this paper may have individual authorship assigned. Where available, figure panel/source data authorship is listed in the following database record: biostudies:S-SCDT-10_1038-S44318-025-00364-w.

## Disclosure and competing interests statement

The authors declare no competing interests.

# Expanded View Figures

▶

**Figure EV1.**  *sur-6* **inactivation affects the duration of mitosis in the AB blastomere.**

(A) Schematic of the first and second cell divisions of *C. elegans* embryos. The arrows show the ingression of the cytokinetic furrows in $P_O$, AB, and $P_1$ blastomeres. The dashed circle line shows nuclear envelope permeabilization. (B) Graphs presenting the duration of interphase (time between furrow ingression in $P_O$ to nuclear envelope permeabilization in AB and $P_1$ blastomeres) and mitosis (time between nuclear envelope permeabilization in AB and P1 to furrow ingression) in wild-type and *sur-6*(sv30) hetero (–/+) and homozygous mutants (-/-). *n* number of embryos analyzed. Non-parametric tests (Kruskal−Wallis) were used to calculate *p* values, which are displayed as follows: ns $= p > 0.05$; * $= p < 0.05$; **** $= p < 0.0001$. Exact *p* values from Interphase AB (L-R); $p < 0.0001$, $p < 0.0001$; Interphase $P_1$ (L-R); $p < 0.0001$, $p < 0.0001$, Mitosis AB (L-R) ; $p < 0.0001$, $p = 0.0116$, Mitosis $P_1$ (L-R); ns. Error bars display the standard error to the mean. ns no-significant differences. (C) Graphs presenting the duration of interphase (time between furrow ingression in $P_O$ to nuclear envelope permeabilization in AB and $P_1$ blastomeres) and mitosis (time between nuclear envelope permeabilization in AB and P1 to furrow ingression) in wild-type and *sur-6ts* mutants. n=number of embryos analyzed. Unpaired *T*-test was used to calculate *p* values, which are displayed as follows: ns $= p > 0.05$; *** $= p < 0.001$; **** $= p < 0.0001$. Exact *p* values from Interphase AB; $p < 0.0001$, Interphase $P_1$; $p < 0.0001$, Mitosis AB; $p < 0.0004$, Mitosis $P_1$ (L-R); ns. Error bars display the standard error to the mean. ns no-significant differences.

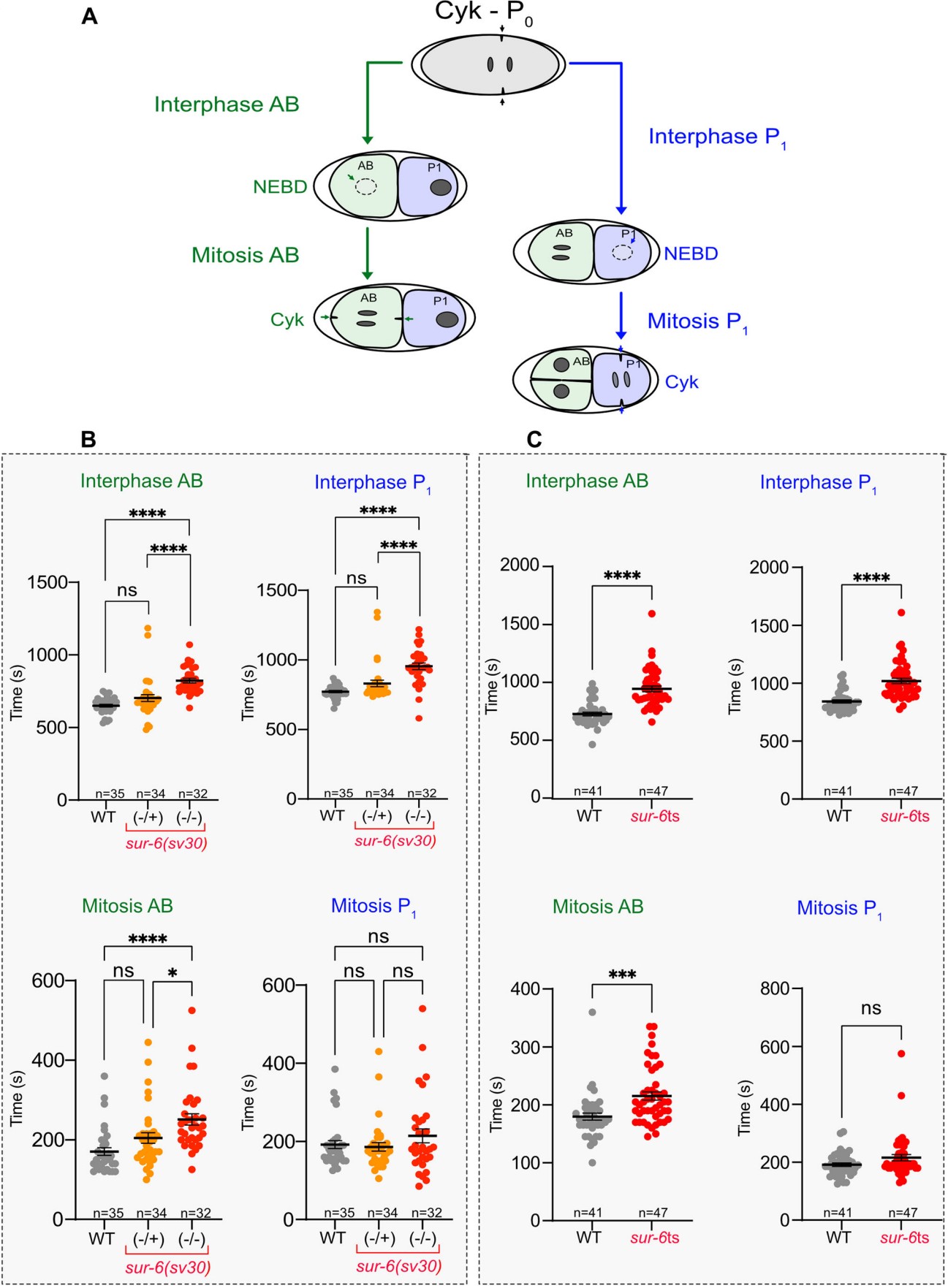

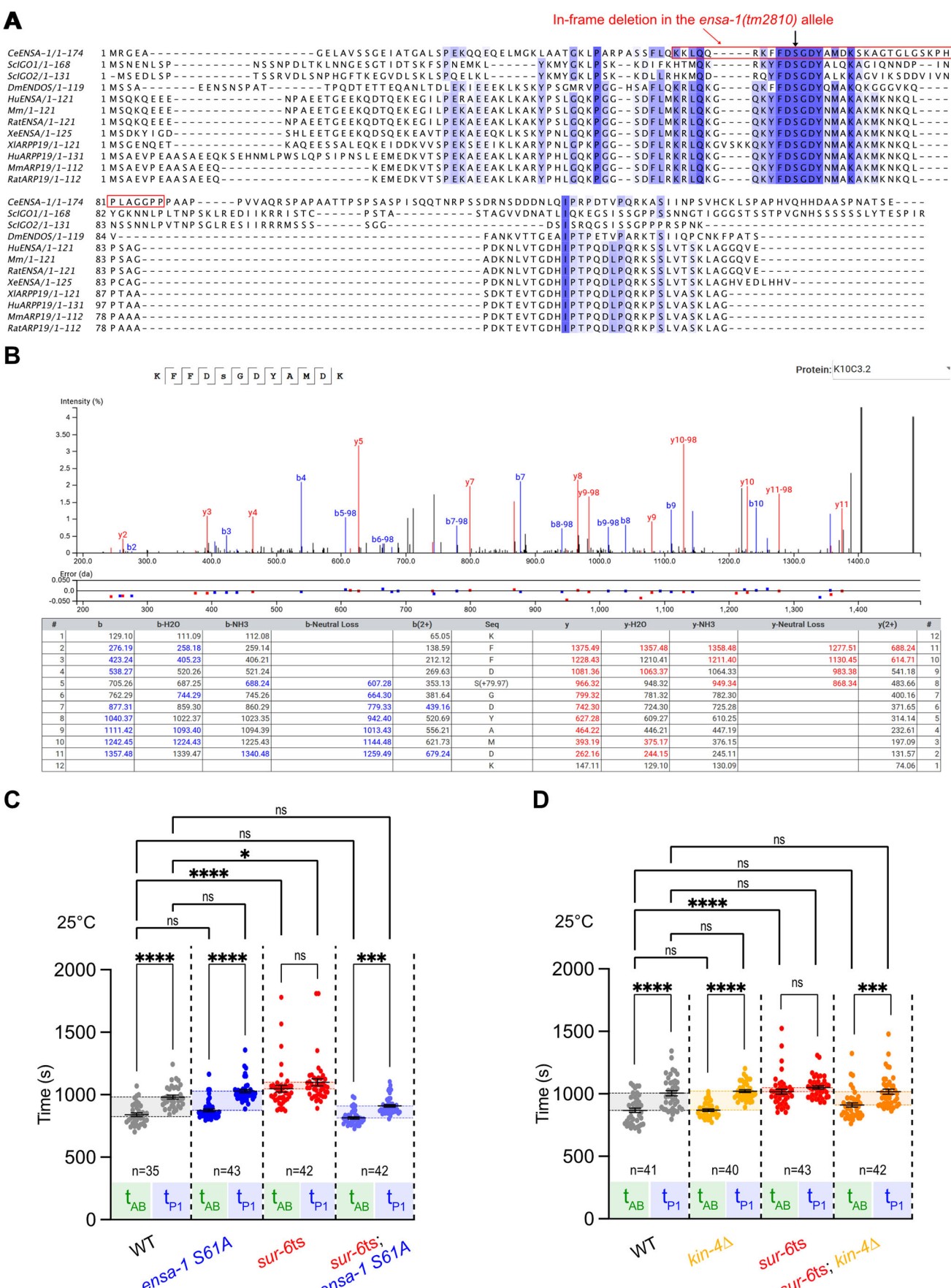

◀

**Figure EV2. KIN-4-dependent ENSA-1 phosphorylation at the DSG sequence motif regulates SUR-6$^{PP2A-B55}$ activity.**

(A) Multiple protein sequence alignments of Endosulfine and Arpp19 from several species (H.s. *Homo sapiens*, X. l, *Xenopus laevis*, M. m. *Mus musculus*, R. n., *Rattus Norvegicus*, D. m. *Drosophila melanogaster*, C. e. *Caenorhabditis elegans, S. c. Saccharomyces cerevisiae*). Endosulfine and Arpp19-related proteins in *S. cerevisiae* are termed Igo1 and Igo2. Note that the DSG sequence motif and surrounding residues are highly evolutionarily conserved. The *ensa-1(tm2810)* allele encodes a truncated protein with an in-frame deletion, removing the DSG sequence motif and surrounding residues. (B) Representative MS/MS spectrum confirming ENSA-1 phosphorylation at site S61. The peptide sequence containing S61 indicates singly charged fragment ions (y + -ion and b + -ion series). The table at the bottom shows the theoretical mass for each fragment ion and the experimentally detected b+ (blue) and y + -ions (red). (C, D) Graph presenting the cell cycle length of AB (light green) and P$_1$ blastomeres (light blue) in embryos of the indicated genotype represented as the mean ± standard error to the mean. *n* number of embryos analyzed. Non-parametric tests (Kruskal−Wallis) were used to calculate *p* values, which are displayed as follows: ns = $p > 0.05$; * = $p < 0.05$; ** = $p < 0.01$; *** = $p < 0.001$; **** = $p < 0.0001$. Exact *p* values from (C) (L-R); $p < 0.0001$, $p < 0.0001$, $p < 0.0001$, $p = 0.0280$, $p = 0.0003$. Exact *p* values from (D) (L-R); $p < 0.0001$, $p < 0.0001$, $p < 0.0001$, $p = 0.0004$. Error bars display the standard error to the mean. ns no-significant differences.

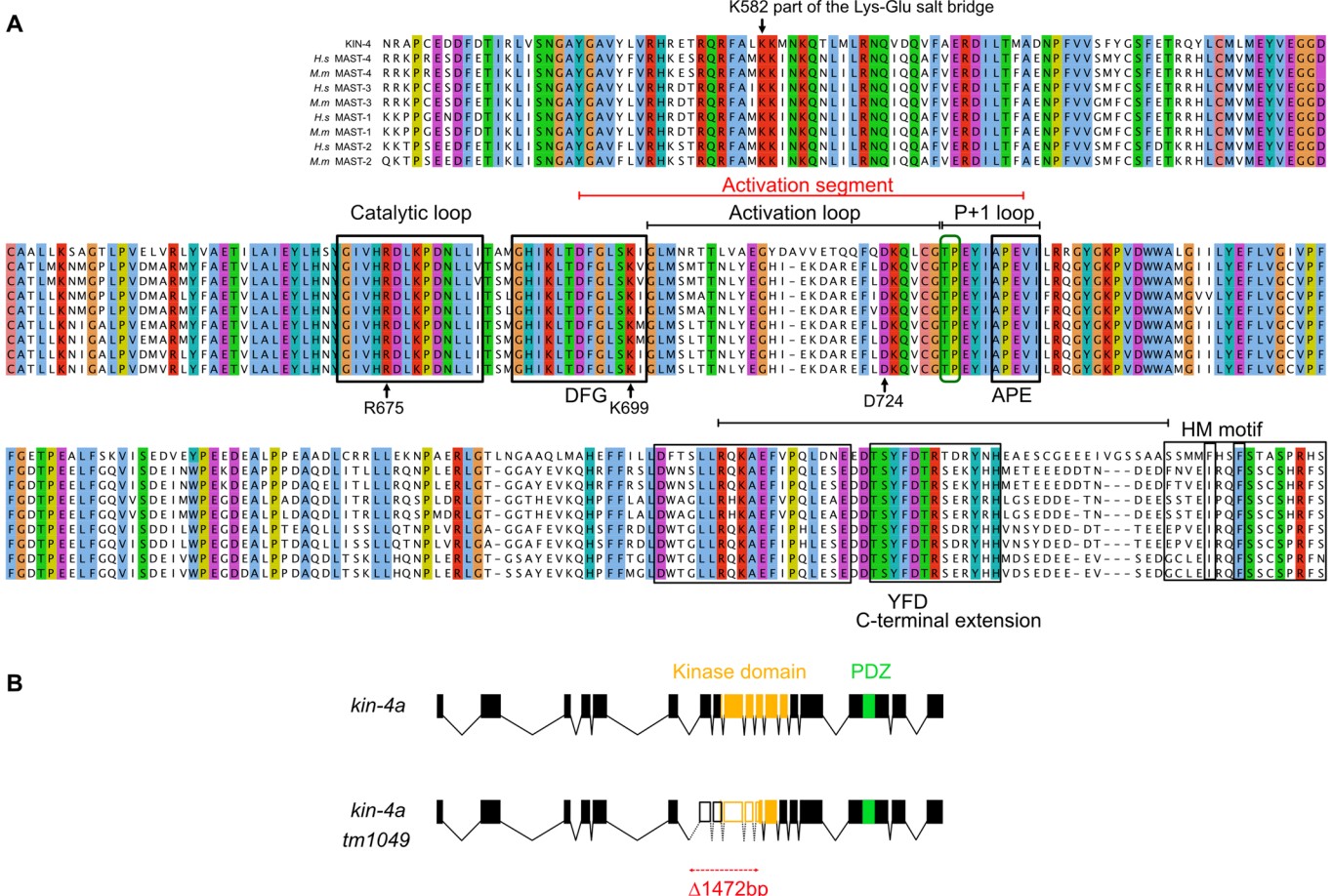

**Figure EV3.   Multiple protein sequence alignments of the MAST kinase domain.**

(A) Protein sequences corresponding to the kinase domains of MAST 1, 2, 3, 4 kinases from *Homo sapiens* (*H. s*), *Mus musculus* (*M. m*) were aligned with the kinase domain of KIN-4 from *Caenorhabditis elegans* (*C. e*). Sequences were aligned using Clustal Omega and visualized with Jalview. The location of the catalytic-, Mg-binding-, P + 1-, activation loops, and C-terminal extension with the hydrophobic motif (HM) are indicated. (B) *kin-4a* gene structure. The *kin-4(tm1049)* allele deletes 1472 bp in the kinase domain (orange).

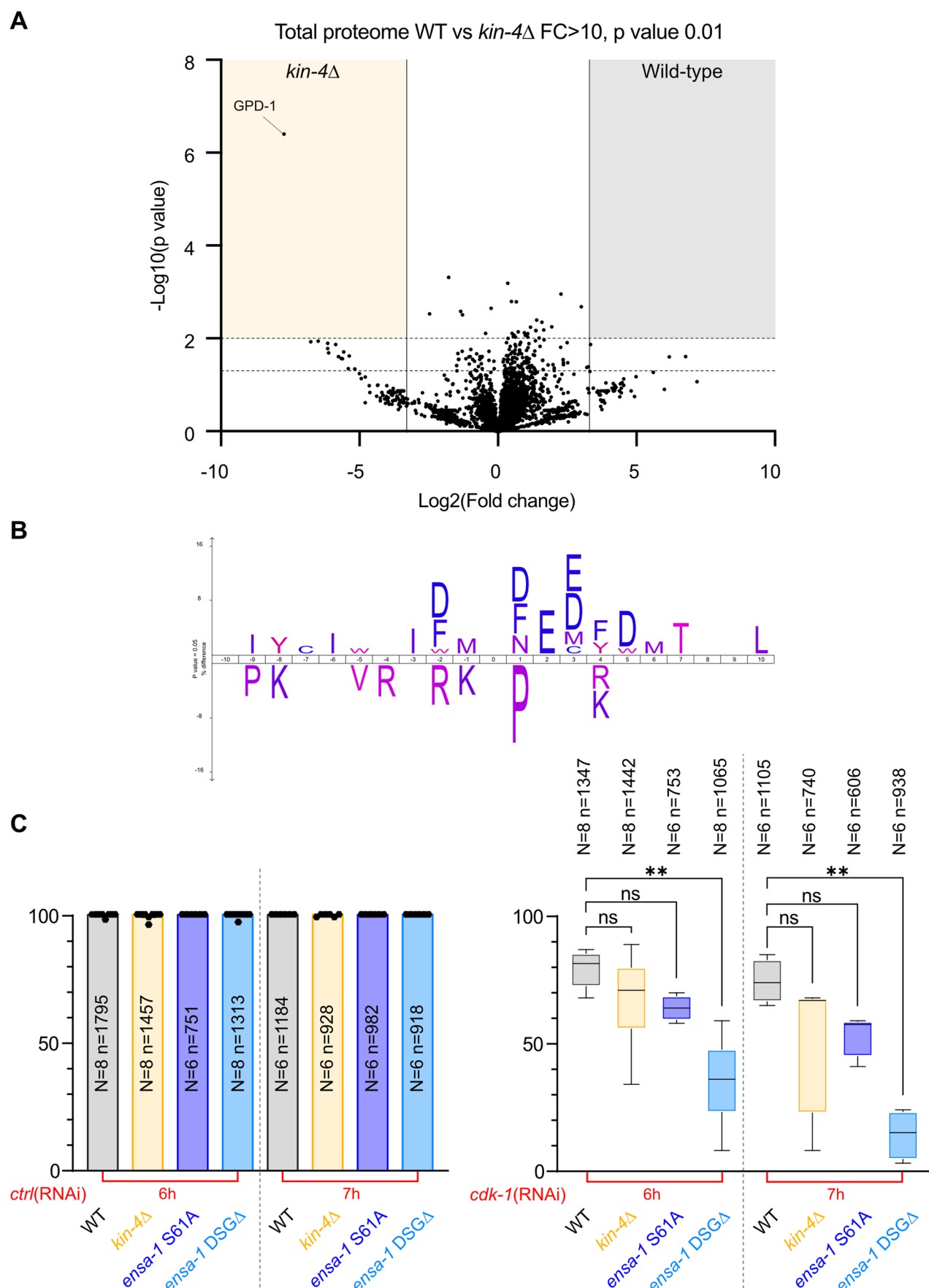

**A** Total proteome WT vs *kin-4Δ* FC>10, p value 0.01

◀ **Figure EV4.** **Quantitative mass spectrometry analysis of wild-type N2 versus *kin-4Δ* proteome.**

(A) Visualization of the quantitative proteomic analysis of wild-type N2 versus *kin-4Δ* in a Volcano plot. Each point on the graph represents a peptide. The $\log_2$-fold change differences between the WT and *kin-4Δ* were plotted on the x-axis, and the $-\log10$ $p$ value differences were plotted on the y-axis. Peptides whose abundance is increased in wild-type versus *kin-4Δ* are located to the right of zero on the x-axis, while peptides whose abundance is decreased are illustrated to the right of zero. Peptides with statistically significant differential abundance lie above the horizontal threshold ($p = 0.01$). The horizontal dashed lines represent a $p$ value of 0.01 and 0.05 (Student's bilateral *t*-test and assuming equal variance between groups, see also methods section), and the vertical dashed lines show a fold change between WT and *kin-4Δ* of 10. (B) IceLogo representation of the phosphopeptide sequences over-represented in *kin-4Δ* strain compared to the total phosphorylated sequences identified in the analysis (Fig. 5B). Significantly over- and under-represented amino acids are visualized. The position 0 corresponds to the position of the phosphorylated serine or threonine. (C) Graph and Box plot showing the percentage of embryonic viability of wild-type N2, *kin-4Δ*, *ensa-1 S61A*, and *ensa-1Δ* exposed to control (Ctrl) or *cdk-1*(RNAi) for 6 or 7 h. $N$ is the number of independent experiments, and $n$ is the total number of embryos counted. A non-parametric test (Kruskal−Wallis) was used to calculate $p$ values displayed as follows: ns = $p > 0.05$; ** = $p < 0.01$, n.s not significant. Exact $p$ values from (L-R) $p = 0.0020$; $p = 0.092$. The box plot indicates the median and interquartile ranges (25th−75th percentile) with whiskers representing min to max values.

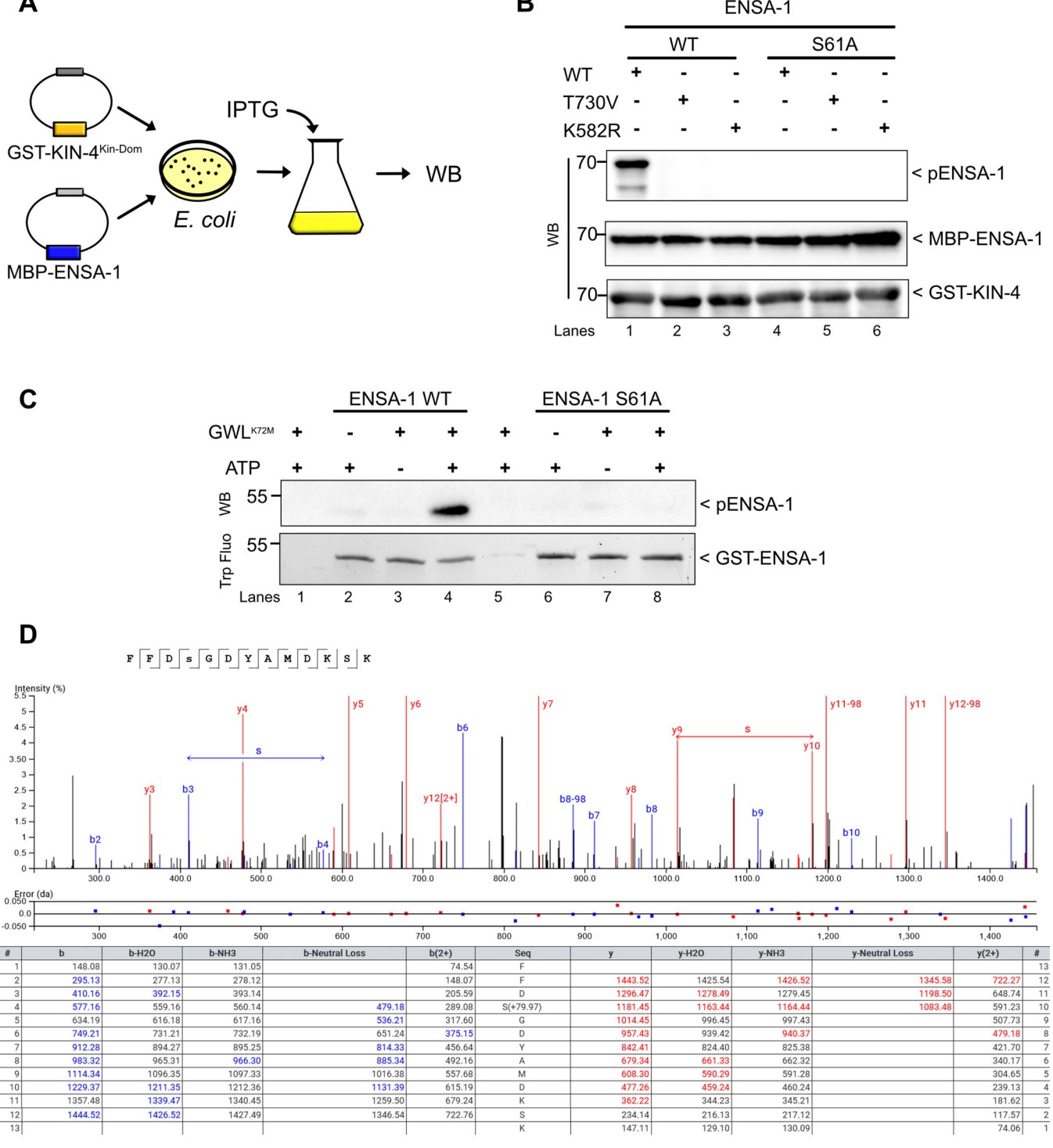

**Figure EV5.  KIN-4 phosphorylates ENSA-1 at the DSG motif in vitro.**

(A) Schematic of the approach to test ENSA-1 phosphorylation by KIN-4^Kin-dom directly in *E. coli*. Plasmids with different replication origins expressing GST-KIN-4^Kin-dom and GST-ENSA-1 wild-type or variants were co-expressed in the *E. coli* BL21 strain (left panel). After protein induction with IPTG, total bacterial lysates were prepared in Laemmli sample buffer, and proteins were separated by SDS-PAGE before transfer on nitrocellulose membrane for western blot analysis using antibodies directed against pENSA-1, MBP, and GST (from top to bottom, right panel). (B) Western blot analysis of kinase reactions was carried out with Xenopus Gwl^K72M and GST-ENSA-1 WT or S61A as substrate. Blots were probed with antibodies to the phospho-DSG motif. The lower panel shows GST-ENSA-1 protein levels detected by tryptophane fluorescence (stain-free, Bio-Rad). (C) Representative MS/MS spectrum confirming ENSA-1 phosphorylation at site S61 site after in vitro phosphorylation by KIN-4. The peptide sequence containing S61 indicates singly charged fragment ions (y + -ion and b + -ion series). (D) The table shows the theoretical mass for each fragment ion and the experimentally detected b+ (blue) and y + -ions (red).

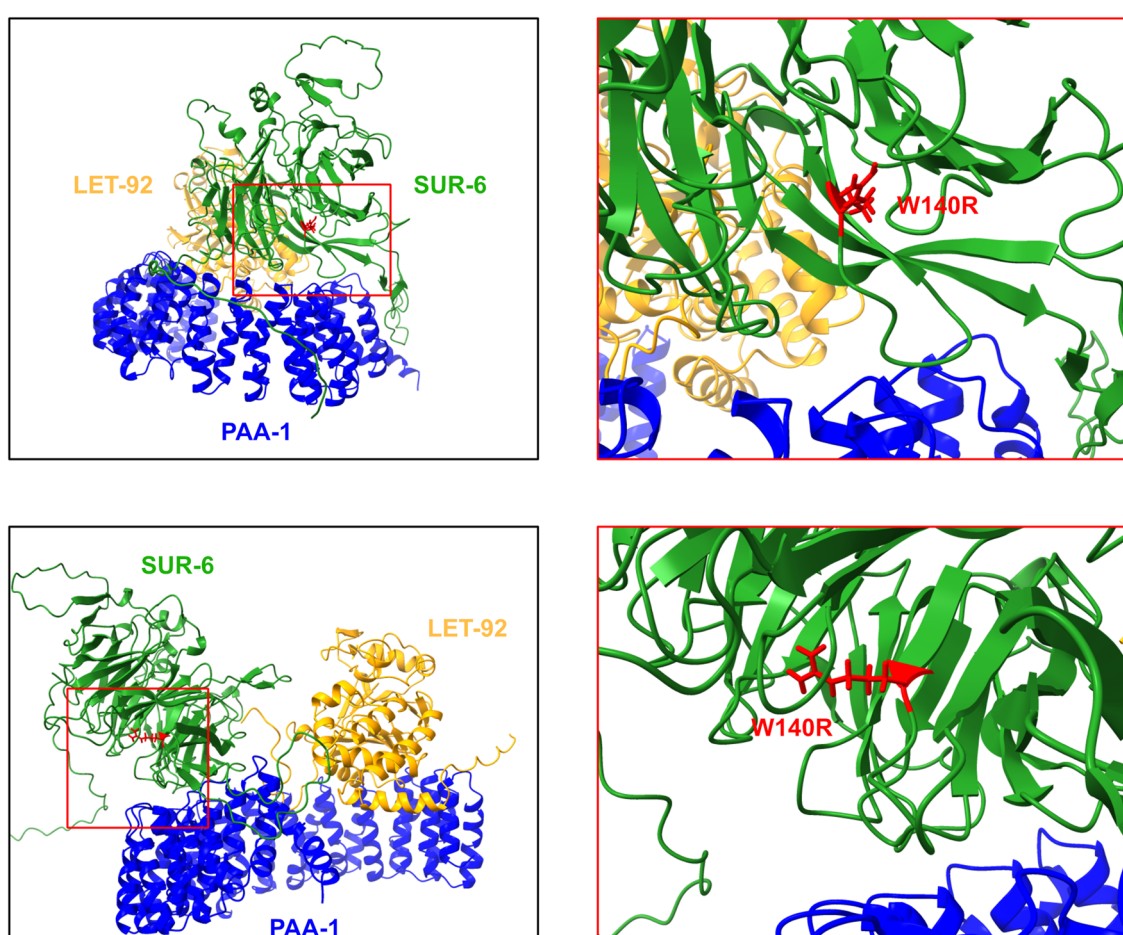

**Figure EV6.  Alphafold model of *C. elegans* PP2A-B55^SUR-6 complex harboring the SUR-6 W140R mutation.**

The worm PP2A-B55 phosphatase complex contains the B55 subunit SUR-6 (green), the scaffold PAA-1 (blue), and the catalytic subunit LET-92 (orange). The Arginine substituting the Tryptophane in position 140, in the *sur-6*ts mutant, is highlighted in red. This residue is located at the interface between SUR-6 and the scaffold PAA-1 subunit and may destabilize the entire complex. Two different orientations and zoomed regions of the PP2A-B55^SUR-6 structure showing the position of the Arginine 140 in red are presented.

