## [Peer Review File · The EMBO Journal]

The MAST kinase KIN-4 carries out mitotic entry functions of Greatwall in *C. elegans*

Ludivine Roumbo, Batool Ossareh-Nazari, Suzanne Vigneron, Ioanna Stefani, Lucie Van Hove, Veronique Legros, Guillaume Chevreux, Benjamin Lacroix, Anna Castro, Nicolas Joly, Thierry Lorca, and Lionel Pintard

Corresponding author(s): Lionel Pintard (lionel.pintard@ijm.fr)

Review Timeline:

Submission Date:	16th Jul 24
Editorial Decision:	9th Aug 24
Revision Received:	8th Nov 24
Editorial Decision:	17th Dec 24
Revision Received:	19th Dec 24
Accepted:	8th Jan 25

Editor: Hartmut Vodermaier

Transaction Report:

Dr. Lionel Pintard
CNRS-UMR7592 / Institut Jacques Monod
Team
Buffon B, CNRS UMR7592
15 rue Helene Brion
Paris 75205
France

9th Aug 2024

Re: EMBOJ-2024-117214
The MAST kinase KIN-4 provides the Greatwall activity in *C. elegans*

Dear Lionel,

Thank you for submitting your study on identification of the Greatwall activity in *C. elegans* to The EMBO Journal. It has now been assessed by three expert referees, whose comments are copied below. As you will see, all reviewers find your study well-planned and the data clear and convincing; but while referees 1 and 2 consider the work also an important contribution to the field, referee 3 is more reserved about its wider significance, and indicated during cross-commenting that deeper insights into KIN-4 regulation (as requested by reviewer 2) would have made the study a stronger candidate.

Faced with these somewhat equivocal recommendations, we concluded that we would on balance be interested in considering a revised version further. However, it would be important to not only satisfactorily address the various specific queries raised by referees 1 and 2, but also to attempt to understand KIN-4 activity control somewhat better, e.g. via approaches such as those proposed by referee 2.

Since our single-major-revision-round policy makes it important to diligently respond to each referee point at the time of resubmission, I would encourage you to contact me with a preliminary point-by-point response already during the early stages of your revision work, in order to clarify how key issues may be addressed and to discuss possible revision plans. We would also be open to extension of the default three-months revision period if needed; our 'scooping protection' (meaning that competing work appearing elsewhere in the meantime will not affect our considerations of your study) would of course remain valid also throughout such an extension.

Detailed information on preparing, formatting and uploading a revised manuscript can be found below and in our Guide to Authors. Thank you again for the opportunity to consider this work for The EMBO Journal, and I look forward to hearing from you in due time.

With kind regards,

Hartmut

- size of the scale bars that are mandatory for all micrograph panels
- the statistical test used to generate error bars and P-values
- the type error bars (e.g., S.E.M., S.D.)
- the number (n) and nature (biological or technical replicate) of independent experiments underlying each data point

- Figures may not include error bars for experiments with $n < 3$; scatter plots showing individual data points should be used instead.

9) To facilitate reproducibility and cross-laboratory adoption of methodologies, please structure the Materials & Methods section as outlined in our guide to authors, including a completed Reagents and Tools Table that can be downloaded from our author guidelines as well (<https://www.embopress.org/page/journal/14602075/authorguide#structuredmethods>).

10) Digital image enhancement is acceptable practice, as long as it accurately represents the original data and conforms to community standards. If a figure has been subjected to significant electronic manipulation, this must be clearly noted in the figure legend and/or the 'Materials and Methods' section. The editors reserve the right to request original versions of figures and the original images that were used to assemble the figure. Finally, we generally encourage uploading of numerical as well as gel/blot image source data; for details see: embopress.org/page/journal/14602075/authorguide#sourcedata

At EMBO Press, we ask authors to provide source data for the main manuscript figures. Our source data coordinator will contact you to discuss which figure panels we would need source data for and will also provide you with helpful tips on how to upload and organize the files.

In the interest of ensuring the conceptual advance provided by the work, we recommend submitting a revision within 3 months (7th Nov 2024). Please discuss the revision progress ahead of this time with the editor if you require more time to complete the revisions. Use the link below to submit your revision:

Link Not Available

Referee #1:

Error-free mitotic progression depends on the precise biochemical interplay between kinases and phosphatases. In metazoans, the onset of mitosis is regulated by the essential mitotic kinase Cdk1-cyclinB, which phosphorylates numerous substrates to drive cells into mitosis. Concurrently, the PP2A-B55 phosphatases that counteract the action of Cdk1-cyclinB on mitotic substrates must be negatively regulated. Greatwall kinases (MASTL in flies) are known to inhibit PP2A-B55 in several metazoan species; however, their identity in *C. elegans* remains unclear. In this study, Roumbo et al. sought to identify and link Greatwall kinase activity to a single MAST kinase, KIN-4, in *C. elegans*. They demonstrate that, similar to higher eukaryotes, KIN-4 in worms phosphorylates ENSA-1 at the DSG motif to control PP2A-B55 activity. Notably, in embryos depleted of ENSA-1 (Fig. 2) or KIN-4 (Fig. 4), the embryonic lethality associated with the partial loss of function of PP2A-B55SUR-6 (*sur-6ts*) is rescued, suggesting that ENSA-1 and KIN-4 negatively regulate B55SUR-6. The authors also show that KIN-4 can functionally replace

Greatwall activity in *Xenopus*. The manuscript is well-written, the experiments are carefully planned, and the findings are potentially significant. Below are a few points, and addressing those will further improve the quality of this work.

Major Points:

In Fig. 1, the authors show that general mitotic progression is delayed in both AB and P1 cells in sv30 and sur-6ts. Here, they calculated the time interval between cytokinetic ingression in P0, AB, and P1 cells. It would be worthwhile to calculate the time interval between P0 cytokinetic ingression and nuclear envelope permeabilization in AB and P1 cells, as well as from nuclear envelope permeabilization to furrow onset in AB and P1 cells. This will help to understand better the mitotic stage where B55SUR-6 activity is crucial.

The authors show that ENSA-1 depletion rescues the loss of embryonic viability upon B55SUR-6 partial loss of function (Fig. 2). They found that multiple phenotypes observed in sur-6ts mutants, such as abnormal PN meeting, DNA segregation errors, and paired nuclei, are rescued upon ENSA-1 depletion. However, there is no mention of the impact of ENSA-1 depletion on the smaller male pronucleus seen in the sur-6ts mutant. Also, why does the percentage of embryos showing paired nuclei vary between Fig. 1H (~18%, 9/51) and Fig. 2E (48%, 11/23)?

The authors suggest that ENSA-1 directly acts on B55SUR-6 (p. 8). Does ENSA-1 depletion enable more complex formation between the PP2A catalytic subunit and B55SUR-6? In other words, is PP2A-B55 activity upregulated upon ENSA-1 depletion?

In Fig. 4, the authors link ENSA-1 phosphorylation with KIN-4 activity, showing that KIN-4 depletion, like ENSA-1 depletion, rescues the embryonic lethality associated with the sur-6 partial loss of function mutant (Fig. 4). In subsequent MS analysis, the authors show that KIN-4 phosphorylates ENSA-1 at the conserved serine 61 residue. In Fig. 3, they used embryonic lysates to test the phosphorylation of S61; however, in Figs. 5A and 5B, they used whole worm extracts to uncover KIN-4 targets. Why wasn't the same experiment conducted using embryonic lysates? Also, do the authors know if replacing S61 with E or D (phosphomimetic amino acids) fails to rescue embryonic viability associated with KIN-4 depletion in the sur-6 ts background?

Based on the phosphopeptide enrichment on page 11, the authors mention that 'inhibitory Y15 phosphorylation was slightly increased in kin-4 mutants, consistent with a reduction in CDK-1 activity'. How do the authors know that CDK-1 activity is reduced in kin-4 mutant animals? If it is reduced, did they observe any phenotypes in the embryos that reflect reduced CDK-1 activity

Minor Points:

It is unclear if the multi-vulvae defects in ENSA-1 or KIN-4 depletion are B55SUR-6-dependent.

In the Introduction, Fig. 1A should be labeled as S1A.

The statement on p. 8, 'Taken together...become toxic when SUR-6 activity is compromised', is confusing.

The amino acid sequence of *C. elegans* ENSA-1 near the DSG motif differs between Fig. 3B and Fig. 5C.

Referee #2:

Since no homolog of Gwl kinase, but of PP2A-B55 and ENSA, has been identified in *C. elegans* it was a longstanding mystery if the pathway of Cdk1/cyclin B-mediated inhibition of PP2-B55 is valid in *C. elegans*. In the submitted ms, Roumbo et al solve this puzzle by identifying KIN-4 as the kinase providing Gwl activity in *C. elegans*. This finding is of high interest for the cell cycle community because it confirms the generality of the Cdk1/cyclin B-mediated PP2-B55 inactivation. The provided data are excellent, the conclusion solid and fully supported by the obtained results. I strongly recommend publication, once the authors have addressed the following points:

- Fig. 1: the authors describe the effects of sur-6 mutations and write about the consequences on cell cycle length of AB and P1 cells, e.g. "..., as in sur-6(sv30) embryos, AB was significantly more affected than P1...". However, this parameter is not directly quantified and statistically analysed in this figure, which the authors could add, e.g. tAB(mut) / tAB(WT) compared to tP1(mut) / tP1(WT)

- The temperature-sensitive mutant of SUR-6 is described to have only partially lost its activity (e.g. start of paragraph describing Fig.2). How do the authors then explain that its effect is equally strong as the effect of the null-mutant in Fig. 1F and 1G?

- Fig. 2E: this reviewer is somewhat puzzled by the results of this experiment. The embryos seem to be totally fine with uncontrolled high (WT + ensa-1 RNAi) and uncontrolled low (sur-6ts + ensa-1 RNAi) SUR-6 phosphatase activity. The embryos only seem to have a problem with low and controlled SUR-6 activity. Is there a possibility to analyze if the level of Cdk1 substrate phosphorylation is altered in these embryos?

- Fig. 4: this reviewer is surprised that constitutive activation of the phosphatase SUR-6 in the kin-4Δ worms seems to result in a strong overall trend for increased phosphorylation of substrates (which is even more surprising considering that Cdk1 activity is decreased). Do the authors have an explanation for this result? Are all the sites enriched in the kin-4Δ mutant non-S/TP sites?

- Fig. 7B: do the authors know the relative amounts of added KIN-4 and Gwl?

- Fig. 7C: the authors write, that they want to test "...whether KIN-4 can substitute for Gwl to maintain the cell cycle arrest of *Xenopus* CSF...". Then (according to the scheme) they perform an experiment in which they first deplete Gwl before they add ectopic KIN-4, i.e. the extract is in interphase (lane 2) when KIN-4 is added, which is basically the same starting position as in Fig. 7B. In my opinion this does not answer if KIN-4 can maintain a *Xenopus* CSF arrest. The question that they want to address would be more appropriately tested by first adding KIN-4 and then depleting Gwl.

- One of the most surprising results of this manuscript is the apparent lack of phospho-regulation of KIN-4. Is there a way to analyze the activity of KIN-4 over a cell cycle? It would be very interesting to know if ENSA-1 phosphorylation changes during the cell cycle.

- Discussion, last paragraph: the authors write "PP2A-55"

Referee #3:

In this study the authors have investigated the Gwl-ENSA-PP2A-B55 pathway in *C. elegans*. *C. elegans* has identifiable orthologues of ENSA and PP2A-B55, but the only AGC kinase gene (Kin-4) does not have the large insertion in the kinase domain characteristic of Gwl in other organisms. Using a combination of genetics, cell biology, and biochemistry, the authors show that Kin-4 functions as expected for Gwl in *C. elegans* and can replace Gwl in *Xenopus* extracts. The experiments in the study are well-controlled and the data are clear and convincing. That said, I am not wholly convinced that this paper has the wide-interest required for publication in *The EMBO Journal*. The results confirm KIN-4 inhibits PP2A-B55 by phosphorylating ENSA on S61 but aside from identifying KIN-4 they do not increase our understanding of the control of mitosis. Thus, the study will primarily be of interest to the *C. elegans* community.

Point-by-point response to the reviewer's comments:

Referee #1:

Error-free mitotic progression depends on the precise biochemical interplay between kinases and phosphatases. In metazoans, the onset of mitosis is regulated by the essential mitotic kinase Cdk1-cyclinB, which phosphorylates numerous substrates to drive cells into mitosis. Concurrently, the PP2A-B55 phosphatases that counteract the action of Cdk1-cyclinB on mitotic substrates must be negatively regulated. Greatwall kinases (MASTL in flies) are known to inhibit PP2A-B55 in several metazoan species; however, their identity in *C. elegans* remains unclear. In this study, Roumbo et al. sought to identify and link Greatwall kinase activity to a single MAST kinase, KIN-4, in *C. elegans*. They demonstrate that, similar to higher eukaryotes, KIN-4 in worms phosphorylates ENSA-1 at the DSG motif to control PP2A-B55 activity. Notably, in embryos depleted of ENSA-1 (Fig. 2) or KIN-4 (Fig. 4), the embryonic lethality associated with the partial loss of function of PP2A-B55SUR-6 (*sur-6ts*) is rescued, suggesting that ENSA-1 and KIN-4 negatively regulate B55SUR-6. The authors also show that KIN-4 can functionally replace Greatwall activity in *Xenopus*. The manuscript is well-written, the experiments are carefully planned, and the findings are potentially significant. Below are a few points, and addressing those will further improve the quality of this work.

Major Points:

In Fig. 1, the authors show that general mitotic progression is delayed in both AB and P1 cells in *sv30* and *sur-6ts*. Here, they calculated the time interval between cytokinetic ingression in P₀, AB, and P1 cells. It would be worthwhile to calculate the time interval between P₀ cytokinetic ingression and nuclear envelope permeabilization in AB and P1 cells, as well as from nuclear envelope permeabilization to furrow onset in AB and P1 cells. This will help to understand better the mitotic stage where B55SUR-6 activity is crucial.

We thank the reviewer for the suggestion. We initially decided to measure the time interval between furrow ingression in P₀, AB, and P₁ cells because these events are easy to score and are non-ambiguous. Scoring nuclear envelope permeabilization by DIC microscopy is less accurate, particularly in *sur-6* mutant embryos, which also present defects in NEBD, as reported recently (Kapoor et al., 2023).

Nevertheless, as suggested, we have now scored the time interval between furrow ingression in P₀ and nuclear envelope permeabilization in AB and P₁ blastomeres (Interphase) and from nuclear envelope permeabilization to furrow onset in AB and P₁ blastomeres (Mitosis).

This analysis shows that interphase in AB and P₁ is extended in *sur-6* mutants. However, the duration of mitosis is mainly delayed in AB but not in the P₁ blastomere. Therefore, the reduction in AB-P1 cell asynchrony primarily results from extended mitosis in *sur-6* mutants. We have included these additional analyses in the revised version of the manuscript in **Fig. S1**.

The authors show that ENSA-1 depletion rescues the loss of embryonic viability upon B55SUR-6 partial loss of function (Fig. 2). They found that multiple phenotypes observed in *sur-6ts* mutants, such as abnormal PN meeting, DNA segregation errors, and paired nuclei, are rescued upon ENSA-1 depletion. However, there is no mention of the impact of ENSA-1 depletion on the smaller male pronucleus seen in the *sur-6ts* mutant.

Here, we used fluorescent histone and tubulin to monitor the position of centrosomes and DNA segregation defects in P₀, AB, and P₁ cells, which is more difficult to score by DIC. Furthermore, we recorded a single focal plane, and most of the embryos recorded started at the time of pronuclear envelope meeting. Therefore, we did not quantify that phenotype.

Nevertheless, we show in Fig. 3F and 4F that *ensa-1* S61A mutation and *kin-4* deletion suppress the small male pronucleus phenotype observed in *sur-6ts* mutants.

Also, why does the percentage of embryos showing paired nuclei vary between Fig. 1H (~18%, 9/51) and Fig. 2E (48%, 11/23)?

We apologize for the confusion. In Fig. 1H, we scored phenotypes of *sur-6ts* embryos by DIC microscopy on animals shifted 5-8 hours at 25°C. In contrast, in Fig. 2E, we scored phenotypes of *sur-6ts* embryos expressing histone and tubulin fused to GFP by spinning disk confocal microscopy on animals shifted at least 48 hours at 25°C. So, the experimental conditions are different.

Moreover, we noticed that the phenotypes of the *sur-6ts* embryos are more severe when histone and tubulin fused to GFP are also expressed. We performed a progeny test and found 10% embryonic lethality in the *sur-6ts* strains versus 42% in the *sur-6ts* expressing histone and tubulin fused to GFP at 16°C. Nevertheless, *ensa-1* depletion can rescue the *sur-6ts* phenotypes.

We have clarified these points in the revised version of the manuscript.

The authors suggest that ENSA-1 directly acts on B55SUR-6 (p. 8). Does ENSA-1 depletion enable more complex formation between the PP2A catalytic subunit and B55SUR-6? In other words, is PP2A-B55 activity upregulated upon ENSA-1 depletion?

PP2A-B55 and PP2A-B56 complexes are present in cells, and so far, no phosphatase inhibitors have been shown to affect complex formation.

Based on the literature on ENSA-1's action mode (Mochida et al., 2010; Gharbi-Ayachi et al., 2010; Labbé et al., 2021; Padi et al., 2024), we thus speculate that PP2A-B55 activity, but not complex formation, is upregulated upon ENSA-1 depletion. We have included these references in the revised version, reminding the reader that Ensa and Arpp19 physically bind PP2A-B55 to inhibit phosphatase activity, as follows:

*This suppression requires residual SUR-6 activity because *ensa-1* depletion had no effect on the lethality of homozygous *sur-6(sv30)* null mutant embryos (Fig. 2D); suggesting that ENSA-1 directly acts on SUR-6 and inhibits PP2A-B55 activity, as reported previously for Arpp19 and Ensa in other systems (Gharbi-Ayachi et al., 2010; Mochida et al., 2010; Labbé et al., 2021; Padi et al., 2024).*

In Fig. 4, the authors link ENSA-1 phosphorylation with KIN-4 activity, showing that KIN-4 depletion, like ENSA-1 depletion, rescues the embryonic lethality associated with the *sur-6* partial loss of function mutant (Fig. 4). In subsequent MS analysis, the authors show that KIN-4 phosphorylates ENSA-1 at the conserved serine 61 residue. In Fig. 3, they used embryonic lysates to test the phosphorylation of S61; however, in Figs. 5A and 5B, they used whole worm extracts to uncover KIN-4 targets. Why wasn't the same experiment conducted using embryonic lysates?

Preparing embryonic lysates for this type of quantitative phosphoproteomic analysis is tedious. Once we noticed that ENSA-1 phosphorylation can be easily detected by tandem mass spectrometry from total worms, and that KIN-4 also regulates SUR-6 activity during vulva development, we proceeded with total worms for the analysis.

Also, do the authors know if replacing S61 with E or D (phosphomimetic amino acids) fails to rescue embryonic viability associated with KIN-4 depletion in the *sur-6ts* background?

We did not do this experiment, but based on the work with *Xenopus* Arpp19, replacing the Serine of the DSG sequence motif with phosphomimetic amino acids (S67D) does not phenocopy the effect of Serine phosphorylation (Dupré et al., 2013).

Based on the phosphopeptide enrichment on page 11, the authors mention that 'inhibitory Y15 phosphorylation was slightly increased in *kin-4* mutants, consistent with a reduction in CDK-1 activity'. How do the authors know that CDK-1 activity is reduced in *kin-4* mutant animals? If it is reduced, did they observe any phenotypes in the embryos that reflect reduced CDK-1 activity.

Previous work in *Xenopus* showed that upon Greatwall and Arpp19 depletion, Y15 of Cdk1 is re-phosphorylated (inhibitory phosphorylation), leading to Cyclin B-Cdk1 inhibition (Vigneron et al., 2009; Gharbi-Ayachi et al., 2010). Consistent with these observations, we observed an increased CDK-1 Y15 phosphorylation upon *kin-4* inactivation in *C. elegans*. However, we agree with the reviewer; we do not provide direct evidence that CDK-1 activity is reduced in *kin-4* mutants. While Y15 phosphorylation is a mark of inhibitory phosphorylation, we do not show any functional consequence on CDK-1 activity.

If CDK-1 activity is diminished in *kin-4* mutants, they might be more sensitive to partial *cdk-1* depletion by RNAi. To test this possibility, we exposed WT, *kin-4* Δ , *ensa-1* Δ , and *ensa-1* S61A animals to partial *cdk-1*(RNAi) conditions and scored embryonic viability.

Although we obtained variable results between experiments performed eight times at two different time points of RNAi (6 and 7 hours), we observed that animals lacking a functional KIN-4/ENSA-1 pathway tend to be more sensitive to partial *cdk-1* inactivation, consistent with reduced CDK-1 activity in these mutants. We have included these data in **Fig. EV4C**.

We consistently noticed that *ensa-1* Δ animals are more sensitive to *cdk-1* partial depletion as compared to *kin-4* Δ and *ensa-1* S61A mutants. This observation suggests that ENSA-1 might have other functions in CDK-1 regulation independently of S61 phosphorylation that will require further investigation.

Minor Points:

It is unclear if the multi-vulvae defects in ENSA-1 or KIN-4 depletion are B55SUR-6-dependent.

This is correct; we only showed that KIN-4, ENSA-1, and SUR-6 have opposing phenotypes on multi-vulvae formation, consistent with KIN-4 and ENSA-1 inhibiting SUR-6 activity.

To test whether the multi-vulvae defects observed upon ENSA-1 or KIN-4 inactivation are B55SUR-6-dependent, we performed double *sur-6* + *ensa-1*(RNAi) or *sur-6* + *kin-4*(RNAi). As shown in the new Fig. 8B, we observed that the double *sur-6* + *ensa-1*(RNAi) or *sur-6* + *kin-4*(RNAi) inactivation phenocopies the single *sur-6* inactivation indicating that the *mov* defects in ENSA-1 or KIN-4 are indeed B55^{SUR-6}-dependent.

In the Introduction, Fig. 1A should be labeled as S1A.

Corrected

The statement on p. 8, 'Taken together...become toxic when SUR-6 activity is compromised', is confusing.

We corrected that sentence. Our data indicate that when PP2A-B55 SUR-6 activity is reduced, the presence of ENSA-1 is detrimental to embryo viability.

We have thus changed the text as follows:

"Taken together, these observations indicate that, although not essential for viability, ENSA-1 is a potent SUR-6^{PP2A-B55} inhibitor, detrimental to embryonic viability when SUR-6^{PP2A-B55} activity is compromised."

The amino acid sequence of *C. elegans* ENSA-1 near the DSG motif differs between Fig. 3B and Fig. 5C.

We thank the reviewer for pointing out this mistake, which has been corrected. The error was in Fig. 3B. We have also included the accession numbers of the aligned proteins in the figure legend.

Referee #2:

Since no homolog of Gwl kinase, but of PP2A-B55 and ENSA, has been identified in *C. elegans* it was a longstanding mystery if the pathway of Cdk1/cyclin B-mediated inhibition of PP2-B55 is valid in *C. elegans*. In the submitted ms, Roumbo et al solve this puzzle by identifying KIN-4 as the kinase providing Gwl activity in *C. elegans*. This finding is of high interest for the cell cycle community because it confirms the generality of the Cdk1/cyclin B-mediated PP2-B55 inactivation. The provided data are excellent, the conclusion solid and fully supported by the obtained results. I strongly recommend publication, once the authors have addressed the following points:

- Fig. 1: the authors describe the effects of *sur-6* mutations and write about the consequences on cell cycle length of AB and P1 cells, e.g. "..., as in *sur-6(sv30)* embryos, AB was significantly more affected than P1...". However, this parameter is not directly quantified and statistically analysed in this figure, which the authors could add, e.g. $tAB(mut) / tAB(WT)$ compared to $tP1(mut) / tP1(WT)$

Thanks for the suggestion. In **Fig. 1** and new **Fig. EV1**, we quantified all the cell cycle stages in wild-type and *sur-6* mutants. We directly compared and statistically analyzed tAB (WT) versus $tAB(mut)$ and did the same for P1 for interphase and mitosis. The graphs **Fig. EV1** indicate that interphase is prolonged in AB and P1 in *sur-6* mutants, but mitosis is delayed in AB and not in P1. We quantified the ratio as suggested by the reviewers, but we can't do a statistical analysis given that we are just comparing ratios of the timing average.

The analysis of the cell cycle timing in wild-type versus *sur-6* mutants presented in **Fig. 1** and **EV1** indicates that the cell cycle is more affected in AB than in P1.

- The temperature-sensitive mutant of SUR-6 is described to have only partially lost its activity (e.g. start of paragraph describing Fig.2). How do the authors then explain that its effect is equally strong as the effect of the null-mutant in Fig. 1F and 1G?

We apologize for the confusion; we needed to be more precise in the text. The phenotypes of the *sur-6* null are more severe than those of the *sur-6ts* embryos. Indeed, we could not even record *sur-6* null embryos at the restrictive temperature of 25°C, and we had to work at 20°C with this mutant. We had this information in the legend of Figure 1, but we have also clarified this point in the text as follows:

*"For *sur-6ts* mutants, we recorded embryos shifted 5-6 hours at 25°C whereas for *sur-6(sv30)* mutants, we had to record embryos at 20°C because they did not divide at 25°C."*

- Fig. 2E: this reviewer is somewhat puzzled by the results of this experiment. The embryos seem to be totally fine with uncontrolled high (WT + *ensa-1* RNAi) and uncontrolled low (*sur-6ts* + *ensa-1* RNAi) SUR-6 phosphatase activity. The embryos only seem to have a problem with low and controlled SUR-6 activity.

Is there a possibility to analyze if the level of Cdk1 substrate phosphorylation is altered in these embryos?

The reviewer is correct. Under unperturbed laboratory conditions with optimal feeding, embryos with "high" (WT + *ensa-1* RNAi) SUR-6 phosphatase activity are fully viable; the cell cycle is only slightly delayed in these embryos, as reported previously by the Goldberg lab

(Kim et al., 2012). Likewise, *sur-6ts + ensa-1(RNAi)* are mainly viable and do not present significant defects. Unfortunately, we do not know what “high” and “low” phosphatase activity means in these conditions. We suspect that the level of Cdk1 phosphorylation is not massively altered; otherwise, we would observe some phenotypes.

As mentioned in our response to the first reviewer, we have tested whether *ensa-1Δ*, *kin-4Δ*, and *ensa-1 S61A* mutants are more sensitive to *cdk-1* depletion, which would be an indication that CDK-1 activity is reduced in these embryos. Although the results are variable, the general trend is that all these mutants are more sensitive to partial *cdk-1* inactivation, which is expected based on the defined role of ENSA-1 and KIN-4 in PP2A-B55 inhibition.

- Fig. 4: this reviewer is surprised that constitutive activation of the phosphatase SUR-6 in the *kin-4Δ* worms seems to result in a strong overall trend for increased phosphorylation of substrates (which is even more surprising considering that Cdk1 activity is decreased). Do the authors have an explanation for this result?

As mentioned in the submitted manuscript, we do not have an explanation for this surprising result. Beyond phosphorylating ENSA-1, the role of KIN-4 is currently unclear, but the sites enriched in the *kin-4Δ* mutant are non-S/TP sites, as mentioned below.

Are all the sites enriched in the *kin-4Δ* mutant non-S/TP sites?

To address this issue, we used IceLogo (Colaert et al., 2009) to compare the sequences of the phosphopeptides enriched in the *kin-4Δ* mutant to the sequence of all the phosphopeptides of the study. This analysis confirmed that the sites enriched in the *kin-4Δ* mutant are non-S/TP sites and are mainly acidic. KIN-4 may thus directly or indirectly regulate the activity of a protein phosphatase dephosphorylating these sites. We have included this analysis in the new **Fig. S4B**.

- Fig. 7B: do the authors know the relative amounts of added KIN-4 and Gwl?

We have included this information in the Materials and Methods section. All the experiments were performed with 23 μ l of *Xenopus* extracts supplemented with 3 μ l of KIN-4 WT [1.43 μ g/ μ l], 3 μ l KIN-4 K582R [3.48 μ g/ μ l], or 3 μ l of hyperactive Gwl K72M [140ng/ μ l].

- Fig. 7C: the authors write, that they want to test "...whether KIN-4 can substitute for Gwl to maintain the cell cycle arrest of *Xenopus* CSF...". Then (according to the scheme) they perform an experiment in which they first deplete Gwl before they add ectopic KIN-4, i.e. the extract is in interphase (lane 2) when KIN-4 is added, which is basically the same starting position as in Fig. 7B. In my opinion this does not answer if KIN-4 can maintain a *Xenopus* CSF arrest. The question that they want to address would be more appropriately tested by first adding KIN-4 and then depleting Gwl.

We agree with the reviewer. Our experiment shows that KIN-4 can restore the CSF status without Greatwall. Therefore, we changed the text to mention that adding KIN-4 to a CSF extract depleted of Gwl can restore the CSF status.

"To corroborate these observations, we asked whether KIN-4 can replace Gwl function during meiosis".

- One of the most surprising results of this manuscript is the apparent lack of phosphoregulation of KIN-4. Is there a way to analyze the activity of KIN-4 over a cell cycle? It would be very interesting to know if ENSA-1 phosphorylation changes during the cell cycle.

We agree with the reviewer; this is one of the most surprising results, but we clearly show that KIN-4, which does not contain a phosphorylatable residue on the T-loop (apparent lack of phosphoregulation), can readily phosphorylate ENSA-1 on the DSG sequence motif. Understanding KIN-4 and MAST kinase regulation is of significant importance, but we believe it is beyond the scope of this study. We already know that human MAST3 readily

phosphorylates *C. elegans* ENSA-1 at the DSG motif, so comparing the substrate specificity and regulation of Gwl and MAST kinases will be very interesting in the future.

Nevertheless, to address the critical point raised by the reviewer, we have tried to use the phospho-ENSA-1 antibody in indirect immunofluorescence experiments on WT or *ensa-1* S61A mutant embryos as a control. Unfortunately, we did not detect any specific signal by this approach. We have also analyzed localization of GFP::ENSA-1, expressed under its endogenous regulatory sequences, in embryos (not shown in the manuscript). As shown in the Figure below, GFP::ENSA-1 is nuclear and cytoplasmic. It does not present a specific enrichment on any cellular structure or on one of the blastomeres, which may explain the difficulties in detecting its phosphorylated forms.

- Discussion, last paragraph: the authors write "PP2A-55"
We thank the reviewer for pointing out this error, which has been corrected.

Referee #3:

In this study the authors have investigated the Gwl-ENSA-PP2A-B55 pathway in *C. elegans*. *C. elegans* has identifiable orthologues of ENSA and PP2A-B55, but the only AGC kinase gene (Kin-4) does not have the large insertion in the kinase domain characteristic of Gwl in other organisms. Using a combination of genetics, cell biology, and biochemistry, the authors show that Kin-4 functions as expected for Gwl in *C. elegans* and can replace Gwl in *Xenopus* extracts. The experiments in the study are well-controlled and the data are clear and convincing. That said, I am not wholly convinced that this paper has the wide-interest required for publication in *The EMBO Journal*. The results confirm KIN-4 inhibits PP2A-B55 by phosphorylating ENSA on S61 but aside from identifying KIN-4 they do not increase our understanding of the control of mitosis. Thus, the study will primarily be of interest to the *C. elegans* community.

We respectfully disagree with the reviewer and believe the manuscript will interest colleagues outside the *C. elegans* community for several reasons.

1) The regulation of cell cycle timing as a function of cell type (or cell identity) during development is an underexplored aspect of cell cycle regulation. Here, using several mutants, we show that the somatic cell cycle of the AB cell is specifically delayed upon PP2A-B55 inactivation. Thus, we show (for the first time) that the PP2A-B55 phosphatase regulates cell cycle duration during animal development.

2) By showing that worm KIN-4 provides the Gwl kinase in *C. elegans*, we also shed light on the evolutionarily conserved MAST kinase family and their apparent lack of phosphorylation, a rarity among kinases. Moreover, when combined with our demonstration that KIN-4 can substitute for Gwl in *Xenopus*, our study reveals that the “important bits” of the kinase structure (at least for its ENSA-regulating function) do not include the split kinase domain, which has been a defining feature of all recognizable Gwl enzymes, but is absent from KIN-4.

Nevertheless, we have modified our abstract to introduce this last point better, which we believe is of interest to a general audience.

References

- Colaert, N., Helsens, K., Martens, L., Vandekerckhove, J., and Gevaert, K. (2009). Improved visualization of protein consensus sequences by iceLogo. *Nat Methods* 6(11), 786-787.
- Dupré, A., Buffin, E., Roustan, C., Nairn, A. C., Jessus, C., and Haccard, O. (2013). The phosphorylation of ARPP19 by Greatwall renders the auto-amplification of MPF independently of PKA in *Xenopus* oocytes. *J Cell Sci* 126(Pt 17), 3916-3926.
- Gharbi-Ayachi, A., Labbe, J. C., Burgess, A., Vigneron, S., Strub, J. M., Brioudes, E., Van-Dorsselaer, A., Castro, A., and Lorca, T. (2010). The substrate of Greatwall kinase, Arpp19, controls mitosis by inhibiting protein phosphatase 2A. *Science* 330(6011), 1673-1677.
- Kapoor, S., Adhikary, K., and Kotak, S. (2023). PP2A-B55^{SUR-6} promotes nuclear envelope breakdown in *C. elegans* embryos. *Cell Rep* 42(12), 113495.
- Kim, M. Y., Bucciarelli, E., Morton, D. G., Williams, B. C., Blake-Hodek, K., Pellacani, C., Von Stetina, J. R., Hu, X., Somma, M. P., Drummond-Barbosa, D., and Goldberg, M. L. (2012). Bypassing the Greatwall-Endosulfine Pathway: Plasticity of a Pivotal Cell-Cycle Regulatory Module in *Drosophila melanogaster* and *Caenorhabditis elegans*. *Genetics* 191(4), 1181-1197.
- Labbé, J. C., Vigneron, S., Méchali, F., Robert, P., Roque, S., Genoud, C., Goguet-Rubio, P., Barthe, P., Labesse, G., Cohen-Gonsaud, M., Castro, A., and Lorca, T. (2021). The study of

the determinants controlling Arpp19 phosphatase-inhibitory activity reveals an Arpp19/PP2A-B55 feedback loop. *Nat Commun* 12(1), 3565.

Mochida, S., Maslen, S. L., Skehel, M., and Hunt, T. (2010). Greatwall phosphorylates an inhibitor of protein phosphatase 2A that is essential for mitosis. *Science* 330(6011), 1670-1673.

Padi, S. K. R., Vos, M. R., Godek, R. J., Fuller, J. R., Kruse, T., Hein, J. B., Nilsson, J., Kelker, M. S., Page, R., and Peti, W. (2024). Cryo-EM structures of PP2A:B55-FAM122A and PP2A:B55-ARPP19. *Nature* 625(7993), 195-203.

Vigneron, S., Brioudes, E., Burgess, A., Labbé, J. C., Lorca, T., and Castro, A. (2009). Greatwall maintains mitosis through regulation of PP2A. *EMBO J* 28(18), 2786-2793.

Dr. Lionel Pintard
CNRS-UMR7592 / Institut Jacques Monod
Team
Buffon B, CNRS UMR7592
15 rue Helene Brion
Paris 75205
France

17th Dec 2024

Re: EMBOJ-2024-117214R
The MAST kinase KIN-4 provides the Greatwall activity in *C. elegans*

Dear Lionel,

Thank you for submitting your revised manuscript to The EMBO Journal, and apologies for the delay in its reevaluation. We have now finally received the below-copied reports of two of the original referees, who were both satisfied with your revisions. We shall therefore be happy to accept the study for publication, pending addressing of a few outstanding editorial issues:

- Please correct the reference list, making sure that for references with more than 10 authors on a paper, only the first 10 should be listed, followed by 'et al.' (please refer to our Guide to Authors for additional information on EMBO J reference format). Also, please adjust the format for citation of preprints as specified in our author guidelines:

The citation in the text should be: "(preprint: NAME1 et al, YEAR)"

The citation in the reference list: "Author NAME1, Author NAME2, ... (YEAR) article title. bioRxiv DOI: XXX"

- For the Source Data, please make sure to upload only a single ZIP archive per main figure, in which with the raw data for individual should be contained in different subfolders.

- Please provide suggestions for a short 'blurb' text prefacing and summing up the study in two sentences (max. 250 characters), followed by 3-5 one-sentence 'bullet points' with brief factual statements of key results of the paper; they will form the basis of an editor-written 'Synopsis' accompanying the online version of the article. Please also upload a synopsis image, which can be used as a "visual title" for the synopsis section of your paper. The image (maybe based on Figure 8D?) should be in PNG or JPG format with the modest dimensions of EXACTLY 550 pixels wide and 300-600 pixels high.

- Finally, I am proposing a few text modifications to title and abstract, to make them somewhat clearer to read, and also to make them more explicit to a broad readership and emphasize broader concepts. Please carefully check these edits in the attached text file, and incorporate them into the final version in case you approve - happy to further discuss if needed!

I am therefore returning the study to you once more, to allow you to incorporate these final changes and upload all requested files. Once we have received them, we shall be able to swiftly proceed with acceptance and production of the manuscript.

With kind regards,

Hartmut

9) To facilitate reproducibility and cross-laboratory adoption of methodologies, please structure the Materials & Methods section as outlined in our guide to authors, including a completed Reagents and Tools Table that can be downloaded from our author guidelines as well (<https://www.embopress.org/page/journal/14602075/authorguide#structuredmethods>).

10) Digital image enhancement is acceptable practice, as long as it accurately represents the original data and conforms to community standards. If a figure has been subjected to significant electronic manipulation, this must be clearly noted in the figure legend and/or the 'Materials and Methods' section. The editors reserve the right to request original versions of figures and the original images that were used to assemble the figure. Finally, we generally encourage uploading of numerical as well as gel/blot image source data; for details see: embopress.org/page/journal/14602075/authorguide#sourcedata

At EMBO Press, we ask authors to provide source data for the main manuscript figures. Our source data coordinator will contact you to discuss which figure panels we would need source data for and will also provide you with helpful tips on how to upload and organize the files.

In the interest of ensuring the conceptual advance provided by the work, we recommend submitting a revision within 3 months (17th Mar 2025). Please discuss the revision progress ahead of this time with the editor if you require more time to complete the revisions. Use the link below to submit your revision:

Link Not Available

Referee #1:

I have gone through the revised manuscript, and I am very much satisfied with the revised version of the manuscript. The authors have thoroughly addressed all of my concerns.

Referee #2:

The authors have fully addressed all my comments/concerns. I therefore recommend the publication of the manuscript.

Dr. Lionel Pintard
CNRS-UMR7592 / Institut Jacques Monod
Team
Buffon B, CNRS UMR7592
15 rue Helene Brion
Paris 75205
France

8th Jan 2025

Re: EMBOJ-2024-117214R1
The MAST kinase KIN-4 carries out mitotic entry functions of Greatwall in *C. elegans*

Dear Lionel,

Thank you for submitting your final revised manuscript for our consideration. I am pleased to inform you that we have now accepted it for publication in The EMBO Journal.

With kind regards,

Hartmut
